# Proximity labeling of axonemal protein CFAP91 identifies EFCAB5 that regulates sperm motility

Haoting Wang [1,2], Keisuke Shimada [1,6], Anh Hoang Pham [1,2], Yuki Oyama [1,2,7], Maki Kamoshita [1,8], Hiroko Kobayashi[1,2,9], Seiya Oura [1,2,10], Norikazu Yabuta[1], Masahito Ikawa [1,2,3,4,5] ✉ & Haruhiko Miyata [1] ✉

Radial spokes (RSs) are conserved multimolecular structures attached to the axonemal microtubule doublets and are essential for the motility control of both cilia and sperm flagella. CFAP91, an RS3 protein, is implicated in human male infertility, yet its molecular function remains poorly understood. Here, we demonstrate that *Cfap91* knockout (KO) mice exhibit impaired sperm flagellum formation and male infertility. Using a transgenic rescue model expressing FLAG- and BioID2-tagged CFAP91, we reveal that CFAP91 immunoprecipitates with RS3 proteins CFAP251 and LRRC23, whose localization is disrupted in *Cfap91* KO sperm flagella. In addition, proximity labeling in mature spermatozoa identifies EFCAB5 as a sperm-specific CFAP91-proximal component. We show that *Efcab5* KO males exhibit reduced sperm motility and fertility. Our findings establish CFAP91 as an essential scaffolder of RS3 assembly and EFCAB5 as a sperm-specialized movement regulator, advancing understanding of axonemal specialization in mammalian spermatozoa and its relevance to male infertility.

Flagella drive the movement of haploid spermatozoa in the female reproductive tract[1], defects of which often lead to male infertility[2,3]. The main component of sperm flagella is the axoneme, a microtubule-based structure composed of two microtubule singlets (central pair) surrounded by nine peripheral microtubule doublets (doublet microtubules)[4] (Fig. 1a). Based on the microtubules, associated proteinaceous structures such as dynein arms, radial spokes (RSs), and microtubule inner proteins are arranged[5]. Noteworthy, the axoneme is not only present in sperm flagella but also in motile cilia. Although both flagellar and ciliary axonemes are thought to share the same origin[6], recent studies have identified that the sperm

axoneme consists of extra structural elements in comparison to the ciliary axoneme[5,7].

RSs are T-shaped structures extended from each of the nine doublet microtubules towards the central pair[8]. Broadly, RSs are critical for motility control in cilia and flagella[9], and defects in the RS structure are linked with male infertility[10–12]. Each of the three RSs, namely RS1, RS2, and RS3, is composed of multiple proteins that form different shapes[13], all of which, can be separated into head, neck, and stalk, according to the spatial localization of each compartment[11,14] (Fig. 1a). Along the doublet microtubules, RS1, RS2, and RS3 are localized repeatedly in the same order in a 96 nm periodicity[15–17].

[1]Research Institute for Microbial Diseases, Osaka University, Suita, Osaka, Japan. [2]Graduate School of Pharmaceutical Sciences, Osaka University, Suita, Osaka, Japan. [3]The Institute of Medical Science, The University of Tokyo, Minato-ku, Tokyo, Japan. [4]Center for Infectious Disease Education and Research, Osaka University, Suita, Osaka, Japan. [5]Center for Advanced Modalities and DDS (CAMaD), Osaka University, Suita, Osaka, Japan. [6]Present address: School of Veterinary Medicine, Rakuno Gakuen University, Ebetsu, Hokkaido, Japan. [7]Present address: Institute of Epigenetics and Stem Cells, Helmholtz Munich, Munich, Bavaria, Germany. [8]Present address: Graduate School of Veterinary Science, Azabu University, Sagamihara, Kanagawa, Japan. [9]Present address: Department of Drug Discovery Medicine, Graduate School of Medicine, Kyoto University, Sakyo-ku, Kyoto, Japan. [10]Present address: Department of Molecular Biology, University of Texas Southwestern Medical Center, Dallas, TX, USA. ✉e-mail: ikawa@biken.osaka-u.ac.jp; hmiya003@biken.osaka-u.ac.jp

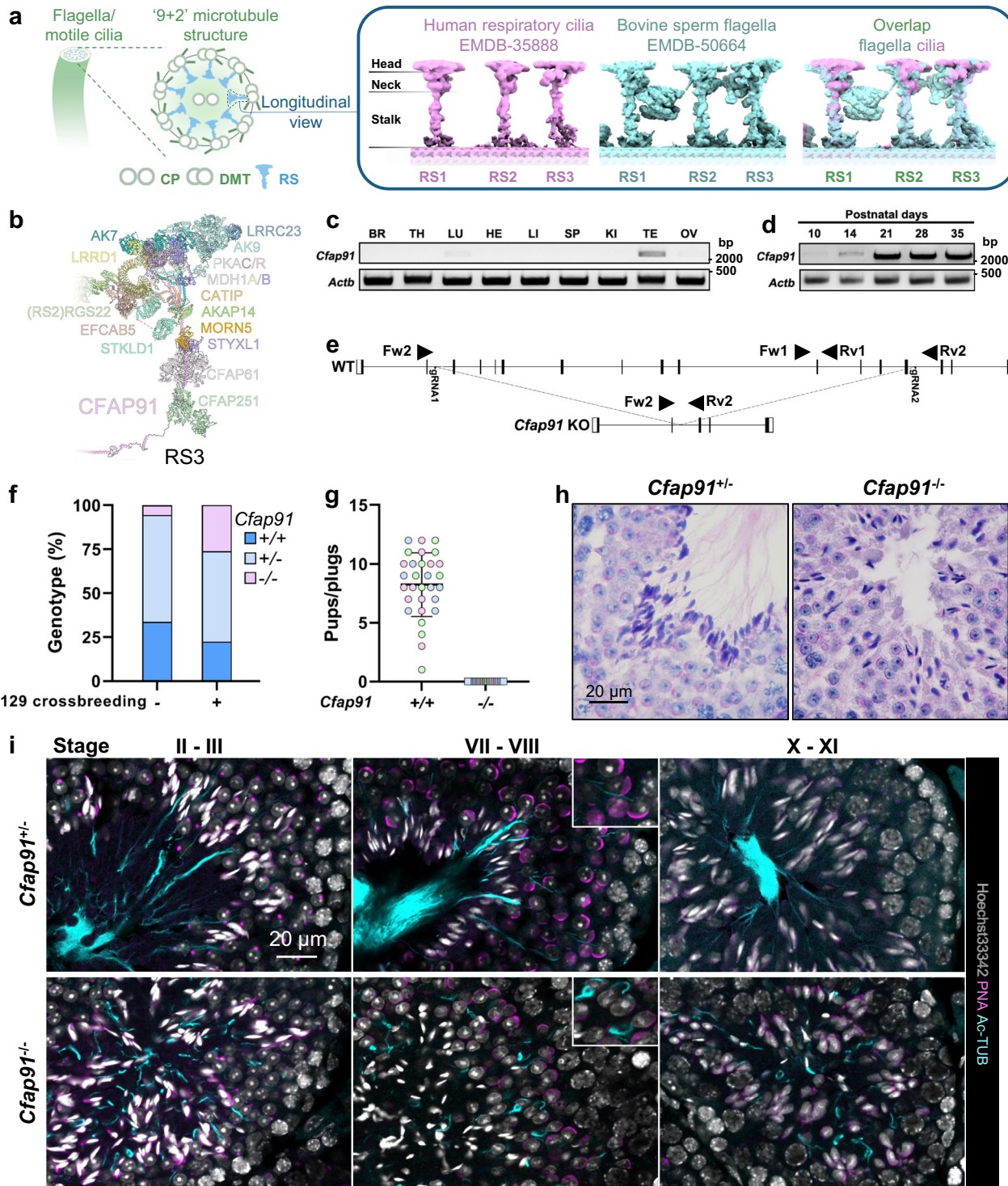

**Fig. 1 | Ablation of *Cfap91* leads to defects in spermiogenesis. a** A schematic diagram of the radial spoke in mouse sperm flagella and human respiratory cilia. CP central pair, DMT doublet microtubule, RS radial spoke. **b** Atomic model of RS3 components. Data was retrieved from the paper by Leung et al.[35]. **c, d** RT-PCR of *Cfap91* utilizing cDNA from multiple mouse organs (**c**) or mouse postnatal testes (**d**). *Actb* was used as a loading control. BR brain, TH thymus, LU lung, HE heart, LI liver, SP spleen, KI kidney, TE testis, OV ovary. **e** A schematic drawing of the KO strategy of *Cfap91*. **f** Ratio of *Cfap91*[+/+], *Cfap91*[+/-], and *Cfap91*[-/-] mice obtained from the mating of *Cfap91*[+/-] mice in the B6D2 or B6D2/129 background. n = 89 for B6D2 background and n = 107 for B6D2/129 background. **g** Fertility tests of *Cfap91*[+/+] and *Cfap91*[-/-] males. Data from each male mouse was individually color-coded. n = 30 plugs examined over 3 males for WT mice, n = 27 plugs examined over 3 males for *Cfap91*[-/-] mice. Data were presented as mean ± SD. **h** Sections of stage VII-VIII seminiferous tubules from *Cfap91*[+/-] and *Cfap91*[-/-] males. **i** Immunohistochemistry of stage II-III, stage VII-VIII, and stage X-XI seminiferous tubules from *Cfap91*[+/-] and *Cfap91*[-/-] males.

Remarkably, RS3 is evolutionarily less conserved between species and structurally diversified between cell types; RS3 is present in *Chlamydomonas reinhardtii* in a truncated form (RS3s)[18]. Moreover, the bridge structure linking RS2 to RS3 is only present in the sperm axoneme but not in the axoneme in other known cell types[7] (Fig. 1a).

During the last phase of spermatogenesis, namely spermiogenesis, axonemes assemble on the centrosome and further attach to the haploid sperm nucleus[19,20]. Spermiogenesis can be divided into 16 steps in mice[21], with axonemes forming during step 2–3[4]. The formation of the axoneme largely depends on intraflagellar transport (IFT), which consists of large protein complexes that are responsible for bidirectional transport in cilia and flagella[22,23]. From step 8, the skirt-like manchette is observed in the caudal part of the sperm nucleus[21], which functions in carving out the base of the sperm nucleus[24]. Anomalies in each of these two processes may lead to male infertility[21,25].

A previous study has identified two biallelic variants of *Cilia and Flagella Associated Protein 91* (*CFAP91*) in male infertile patients, which led to severe astheno-teratozoospermia, a condition that is accompanied by diminished sperm motility and abnormal sperm morphology[26]. The underlying mechanism of CFAP91 in regulating sperm flagellum biogenesis is not fully understood. According to the TreeFam database[27], *Cfap91* is conserved in 84 species out of 109 eukaryotes and is present in all mammals (40/40) (Supplementary Fig. 1a). Equally important, CFAP91 is also conserved in humans and major experimental animals (*C. reinhardtii*, *Capra hircus*, *Drosophila melanogaster*, *Danio rerio*, *Mus musculus*) (Supplementary Fig. 1b). In *C. reinhardtii*, FAP91 (ortholog of CFAP91) was found to form a complex with FAP251 (ortholog of CFAP251)[28,29]. This CFAP91-CFAP251 complex is found to be involved in the correct formation of RS3 in *Tetrahymena thermophila*[29,30]. Recently, by integrating artificial intelligence modeling and cryo-electron microscopy/tomography, CFAP91 has been shown to localize in RS3 and bind with CFAP251 in mouse ependymal cilia and human respiratory cilia[31,32]. Notably, according to a high-resolution study published recently, CFAP91 has been shown to exhibit a distinctive localization in bovine sperm axoneme: the N-terminal region of CFAP91 is localized from the RS2 base to the RS3 base through the base plate of the nexin-dynein regulatory complex (N-DRC) and is intertwined with the protofilament formed by CCDC39/40[33,34]. Meanwhile, the C-terminal region extends from the RS3 base to the RS3 neck[35] (Fig. 1b). In the present work, by recruiting cell biological, proteomics, and genetic techniques, we revealed that depletion of CFAP91 led to abnormal recruitment of its binding partners and resulted in impaired sperm tail elongation. Moreover, we employed proximity labeling in mature spermatozoa and identified EFCAB5 as a sperm-specific CFAP91-proximal protein, in which deletion affected sperm motility.

## Results
### Generation of *Cfap91* KO mice
To elucidate the organic expression profile of *Cfap91*, reverse transcription polymerase chain reaction (RT-PCR) was performed on multiple adult mouse tissues. *Cfap91* exhibited a predominant expression in mouse testes with a weak expression in the lungs (Fig. 1c). RT-PCR using mouse testicular tissues from postnatal day 10 to day 35, demonstrated that *Cfap91* started to express in mouse testes from postnatal day 14 (Fig. 1d). This period corresponds to the first presence of spermatocytes during the first wave of spermatogenesis[36].

To understand the role of *Cfap91* in spermatogenesis, we generated *Cfap91* knockout (KO) mice. We electroporated Cas9 and two guide RNAs (gRNAs) that targeted the genomic region of *Cfap91* into 64 zygotes obtained from mating (B6D2F1 x B6D2F1) (Fig. 1e). Next, we transferred 62 two-cell embryos into the oviduct of two pseudopregnant females and obtained 8 pups; 6 of which possess a large deletion

in *Cfap91*. Subsequently, the *Cfap91* mutant F0 mouse was mated with the B6D2F1 wild type (WT) mouse, which produced *Cfap91*[+/-] F1 mice; *Cfap91*[+/-] F1 mice were then caged with each other to obtain *Cfap91*[-/-] mice. Later, we confirmed there were 32,204 base pairs (bps) of deletion and 4 bps of insertion in the genomic region of *Cfap91* (Supplementary Fig. 1c). This deletion was confirmed by genomic PCR (Supplementary Fig. 1d). Similar to the KO studies of some axonemal proteins[1], we could not obtain *Cfap91*[-/-] mice according to the Mendelian ratio, only 5.6% of the pups were *Cfap91*[-/-] mice in the *Cfap91* heterozygous mating of the B6D2 background (5/89 pups) (Fig. 1f). Even then, they all died before 8 weeks of age. As mutations in other *Cfap* genes were reported to cause hydrocephalus[37,38], we hypothesized that *Cfap91*[-/-] mice in the B6D2 background may also carry hydrocephalus. An earlier report established that mice in the 129S6/SvEvTac (129) background have less susceptibility to hydrocephalus[39]. Therefore, we crossed B6D2 *Cfap91*[+/-] mice with 129 background WT mice. Mice produced from this mating are defined as B6D2/129 background mice. We caged B6D2/129 *Cfap91*[+/-] male and female in 3 individual cages and were able to obtain *Cfap91*[-/-] mice under the Mendelian ratio (28/107 pups, 26.2%) (Fig. 1f). Afterwards, we used the mice in the B6D2/129 background for further experiments.

### Ablation of *Cfap91* resulted in male infertility, accompanied by defective spermiogenesis
To test the fertility, we caged *Cfap91*[-/-] males with WT females and found that *Cfap91*[-/-] males were infertile in 3 trials (Fig. 1g). To discover why *Cfap91*[-/-] males were infertile, we first performed a gross inspection of their testes. Although there was no alteration in their body weight when compared with *Cfap91*[+/-] males (Supplementary Fig. 2a), the weight of *Cfap91*[-/-] testes was significantly decreased (Supplementary Fig. 2b, c). We then analyzed the cross-sections of *Cfap91*[-/-] testes. It should be noted that sections of seminiferous tubules can be categorized into 12 stages in mice according to their histology[40]. In stage VII-VIII, elongated spermatids are localized near the center of the seminiferous tubules, with tails aligning in the center of the lumens. In contrast, we could not find the sperm tails in the seminiferous tubules of *Cfap91* KO males (Fig. 1h). To inspect sperm tail biogenesis in *Cfap91*[-/-] testes, we performed immunohistochemistry (IHC) on testis sections with an anti-acetylated tubulin antibody and found shorter sperm tails in *Cfap91*[-/-] mice (Fig. 1i). Furthermore, morphology of round spermatids was comparable between *Cfap91*[+/-] males and *Cfap91*[-/-] males, whereas elongating/elongated spermatids showed abnormal head morphology in *Cfap91*[-/-] males (Supplementary Fig. 2d). These data suggest that *Cfap91* KO males undergo defective spermiogenesis.

### *Cfap91* KO males showed oligo-astheno-teratozoospermia
We then examined the epididymis and cauda epididymal spermatozoa of *Cfap91*[-/-] males. It showed that *Cfap91*[-/-] males had a smaller and significantly lighter epididymis (Supplementary Fig. 3a, b). When the cauda epididymal section was inspected, only a few sperm heads were found in *Cfap91*[-/-] males (Supplementary Fig. 3c). Consistently, fewer number of spermatozoa were found in the cauda epididymis of *Cfap91*[-/-] males (Supplementary Fig. 3d). Furthermore, spermatozoa from *Cfap91*[-/-] cauda epididymis showed abnormal morphology (Fig. 2a), including shorter tails (Fig. 2b) and abnormal heads (Fig. 2c). In addition, *Cfap91* KO spermatozoa were completely immotile (Fig. 2d).

Since motile cilia also possess axonemes with radial spokes, we examined ependymal and tracheal motile cilia in *Cfap91*[-/-] mice (B6D2/129 background), using an anti-acetylated tubulin antibody. However, there were no apparent differences found between *Cfap91*[+/-] and *Cfap91*[-/-] mice (Supplementary Fig. 3e), indicating that ciliary elongation was endurable in *Cfap91*[-/-] mice in contrast to disrupted sperm flagellum formation.

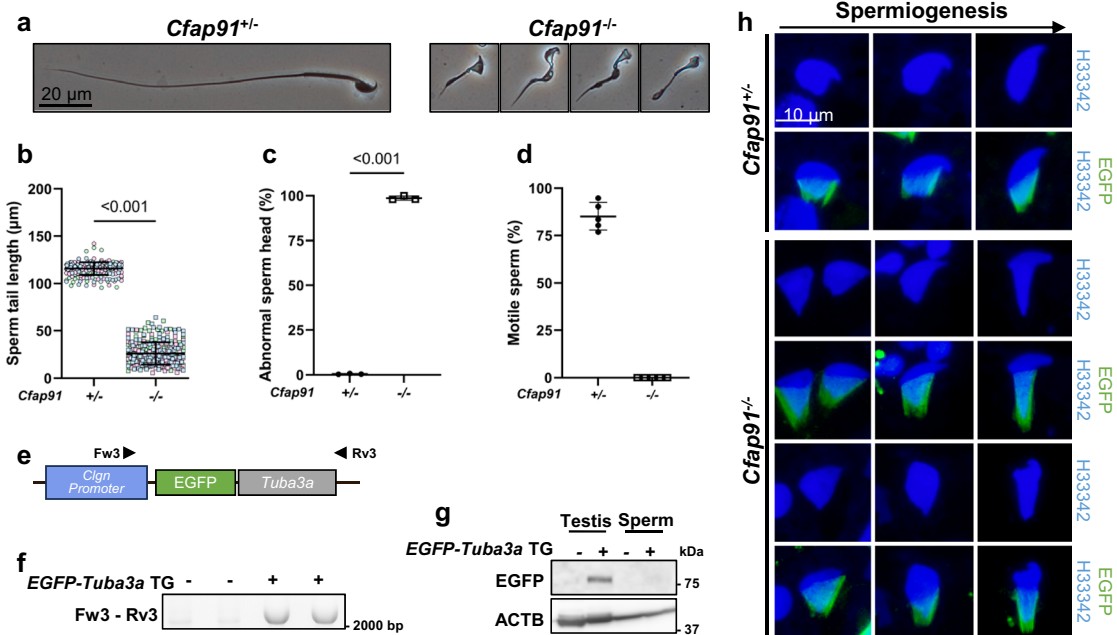

**Fig. 2 | *Cfap91⁻/⁻* males show oligo-astheno-teratozoospermia. a** Phase contrast images of cauda epididymal spermatozoa from *Cfap91⁺/⁻* and *Cfap91⁻/⁻* males. **b** Sperm tail length of cauda epididymal spermatozoa from *Cfap91⁺/⁻* and *Cfap91⁻/⁻* males. Data from each male mouse was individually color-coded. n = 267 spermatozoa examined over 3 males for *Cfap91⁺/⁻* mice, n = 300 spermatozoa examined over 3 males for *Cfap91⁻/⁻* mice. Data were presented as mean ± SD. An unpaired two-tailed t-test was performed for statistical analysis. P < 1.0E-15. **c** Ratio of the abnormal head of cauda epididymal spermatozoa from *Cfap91⁺/⁻* and *Cfap91⁻/⁻* males (n = 3 males for each genotype). Data were presented as mean ± SD. An unpaired two-tailed t-test was performed for statistical analysis. P = 1.5E-8. **d** Ratio of motile spermatozoa from the cauda epididymis of *Cfap91⁺/⁻* and *Cfap91⁻/⁻* males. *Cfap91⁻/⁻* males showed no motile spermatozoa (n = 5 males for each genotype). Data were presented as mean ± SD. **e, f** A schematic diagram of the *EGFP-Tuba3a* transgene driven by a *Clgn* promoter (**e**). The results of genomic PCR performed with the corresponding primers are shown (**f**). **g** Immunoblotting of EGFP in testes and cauda epididymal spermatozoa of WT and *EGFP-Tuba3a* TG males. ACTB was used as a loading control. **h** Imaging of manchette using *EGFP-Tuba3a* TG mice with *Cfap91⁺/⁻* and *Cfap91⁻/⁻* genotypes. Hoechst 33342 (H33342) was used to visualize the nuclei of spermatids. Images are arranged from left to right with the progression of spermiogenesis.

As *Cfap91* KO spermatozoa showed abnormal sperm head morphology, we attempted to visualize the morphological change of sperm heads together with the manchette. As the manchette mainly consists of TUBA3 (alpha-tubulin) and TUBB4 (beta-tubulin) among tubulin proteins[41], we generated a transgenic (TG) mouse expressing TUBA3A fused with an N-terminal EGFP under testis-specific *Clgn* promoter (Fig. 2e). We confirmed that the transgene was incorporated into the genome by genomic PCR (Fig. 2f), and confirmed the expression of EGFP-TUBA3A by immunoblotting in testes but not in cauda epididymal spermatozoa lacking the manchette (Fig. 2g). We then crossed *EGFP-Tuba3a* TG mice with *Cfap91* mutant mice to analyze the manchette-dependent morphological changes of sperm heads in *Cfap91⁻/⁻* males (Fig. 2h). For early elongating spermatids, no overt abnormalities were found in manchettes and sperm heads of *Cfap91⁻/⁻* mice. However, in later steps, *Cfap91⁻/⁻* males showed longer manchettes, accompanied by abnormal head morphology, indicating that disrupted sperm head morphogenesis was likely caused by abnormally elongated manchettes.

We then applied transmission electron microscopy (TEM) on testes and cauda epididymis to observe ultrastructural defects. In testis samples, *Cfap91* KO spermatids showed defective microtubule structures (Fig. 3a–c). Cauda epididymal spermatozoa of *Cfap91⁻/⁻* males also showed disordered axoneme structures (Fig. 3d–h). In addition, abnormalities were found in the mitochondrial sheath, outer dense fiber, and fibrous sheath in *Cfap91* KO spermatozoa (Fig. 3d–h). In late spermiogenesis, the electron-dense annulus structure migrates distally away from the sperm head, which marks the boundary of the midpiece and principal piece, and allows the formation of mitochondrial sheath[42]. However, in cauda epididymal spermatozoa from *Cfap91⁻/⁻* males, the annulus could not migrate far enough to the distal

region (Fig. 3h). Overall, our results indicate that spermiogenesis was abnormal in *Cfap91⁻/⁻* males, which resulted in oligo-astheno-teratozoospermia.

### *Cfap91* transgene was able to rescue the fertility of *Cfap91⁻/⁻* males

To confirm whether the infertility found in *Cfap91⁻/⁻* males was caused by the absence of *Cfap91* but not by other factors, we generated *Cfap91* TG mice. We tagged BioID2[43] and 3×FLAG in the C-terminus of CFAP91, where its expression was regulated by a testis-specific *Clgn* promoter (Fig. 4a). Hereby, mice carrying this transgene are referred to as *Cfap91* TG mice. The genotype of *Cfap91* TG mice was confirmed by genomic PCR (Supplementary Fig. 4a) and the protein expression of CFAP91-BioID2-3×FLAG was validated by immunoblotting (Fig. 4b). We then found that *Cfap91⁻/⁻* TG males showed a comparable sperm tail length, motile sperm rate, abnormal sperm head rate, and sperm count to *Cfap91⁺/⁻* males (Supplementary Fig. 4b–f). On top of that, *Cfap91⁻/⁻* TG males showed similar fertility when compared with WT males (Fig. 4c). These results indicate that the oligo-astheno-teratozoospermia of *Cfap91⁻/⁻* mice is indeed caused by the absence of CFAP91.

### CFAP91 is localized in sperm tails and fractionated with axonemal proteins

To examine the localization of CFAP91, we performed IHC using an anti-FLAG antibody to detect CFAP91-BioID2-3×FLAG in *Cfap91⁻/⁻* TG males. In testicular sections, CFAP91 signals were found in sperm tails (Fig. 4d). To better understand the spatiotemporal localization of CFAP91, we isolated germ cells from the testis and performed immunocytochemistry (ICC). We found that CFAP91 localizes in both sperm tail and cytoplasm during the round spermatid phase, with signals in

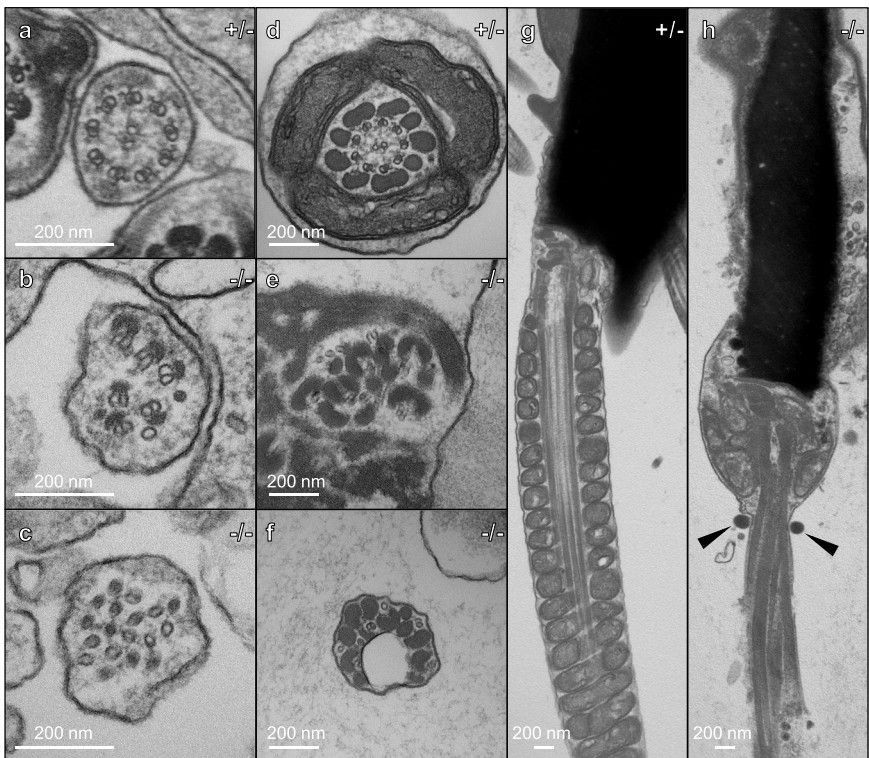

**Fig. 3 | Ultrastructural defects in *Cfap91* KO spermatozoa. a–f** Cross-sections of spermatids from the testes (**a–c**), and spermatozoa in the cauda epididymis (**d–f**) of *Cfap91*⁺/⁻ and *Cfap91*⁻/⁻ males. **g, h** Longitudinal sections of spermatozoa in the cauda epididymis of *Cfap91*⁺/⁻ and *Cfap91*⁻/⁻ males. Genotypes of *Cfap91* are indicated on the upper right of each figure. Abnormal proximal annulus found in *Cfap91* KO spermatozoa is shown with black triangles.

the cytoplasm being largely diminished when entering the elongating spermatid phase (Fig. 4e). This localization pattern suggests that cytoplasmic CFAP91 is likely transported to sperm tails during spermiogenesis. CFAP91 signals were then found in whole tails of mature spermatozoa collected from *Cfap91*⁻/⁻ TG cauda epididymis (Supplementary Fig. 4g). To confirm our findings, we performed immunoblotting on sperm head-tail separated lysates from *Cfap91*⁻/⁻ TG mice. CFAP91 was detected in the tail lysate but not in the head lysate (Fig. 4f). We further questioned which compartment CFAP91 localizes to in sperm tails. We therefore performed sperm fractionation, in which sperm tails were fractionated into three compartments mainly containing transmembrane and cytosolic proteins, axonemal proteins, and outer dense fiber/fibrous sheath proteins (Triton-soluble, SDS-soluble, and SDS-resistant compartments, respectively)[3,44]. CFAP91 showed signals in the SDS-soluble compartment (Fig. 4g), suggesting that CFAP91 may be an axoneme-associated protein in spermatozoa.

## CFAP91 immunoprecipitates with RS3 proteins during spermiogenesis

To explore the relationship between CFAP91 and other proteins, we performed immunoprecipitation-mass spectrometry (IP-MS) on the testis lysate of *Cfap91*⁻/⁻ TG males using an anti-FLAG antibody (Supplementary Fig. 5a). We found that 45 proteins were significantly enriched in the CFAP91 immunoprecipitates (Fig. 5a and Supplementary Data 1). Among the immunoprecipitates, seven proteins, including CFAP91, AK7, AK9, CATIP, CFAP251, LRRC23 and MDH1B, have been shown to localize in RS3 of mammalian spermatozoa (Fig. 5a and Supplementary Data 2), consistent with CFAP91 localization in sperm RS3[35]. We confirmed the presence of CFAP251 and LRRC23 in the CFAP91 immunoprecipitates with immunoblotting (Fig. 5b). Since the anti-LRRC23 antibody works for IP, association of CFAP91 and LRRC23 in testes was further confirmed by performing IP and subsequent

immunoblotting for CFAP91-3xFLAG (Fig. 5c). During spermiogenesis, CFAP91 co-localizes with CFAP251 and LRRC23 in sperm tails (Supplementary Fig. 5b). To test if this CFAP91-CFAP251/LRRC23 complex is also present in the early stage of spermatogenesis, we performed co-IP on postnatal 15 days (P15) testes, before the appearance of haploid spermatids with tails[36]. Intriguingly, CFAP251 but not LRRC23 immunoprecipitated with CFAP91 in P15 testes (Fig. 5d and Supplementary Fig. 5c), suggesting that the CFAP91-CFAP251 complex is formed first during spermatogenesis.

We next questioned how the absence of *Cfap91* impacted its immunoprecipitates and applied IHC on *Cfap91*⁻/⁻ testis sections. We demonstrated that CFAP251 and LRRC23 could not localize in sperm tails during spermiogenesis in *Cfap91*⁻/⁻ males (Fig. 5e). Similarly, the protein levels of CFAP251 and LRRC23 showed no alteration in *Cfap91*⁻/⁻ testes in immunoblotting; however, they were downregulated in *Cfap91* KO spermatozoa (Fig. 5f). In addition, IFT140, which is responsible for sperm tail elongation, was not present in WT epididymal spermatozoa[45], but IFT140 remained in the spermatozoa of *Cfap91*⁻/⁻ males (Fig. 5f), suggesting that sperm tail formation may halt in the absence of CFAP91. Overall, these data suggested that the localization of CFAP91-associating RS3 proteins was disrupted in sperm tails of *Cfap91*⁻/⁻ males.

## CFAP91 immunoprecipitates with BBSome proteins during spermiogenesis

In addition to RS3 proteins, 5 members of the Bardet-Biedl syndrome (BBS) protein family, BBS1, BBS2, BBS5, BBS7 and BBS9, were detected in CFAP91 immunoprecipitates (Fig. 5a, b). Previous reports indicated that BBS1/2/4/5/7/8/9/18 form a complex called BBSome and function in protein transport in cilia/flagella[46]. KO of individual members of BBSome has been shown to cause the shortening of sperm flagella in mice[47–51]. When comparing with our previous IP-MS results of MYCB-PAP, a central pair apparatus protein essential for sperm tail

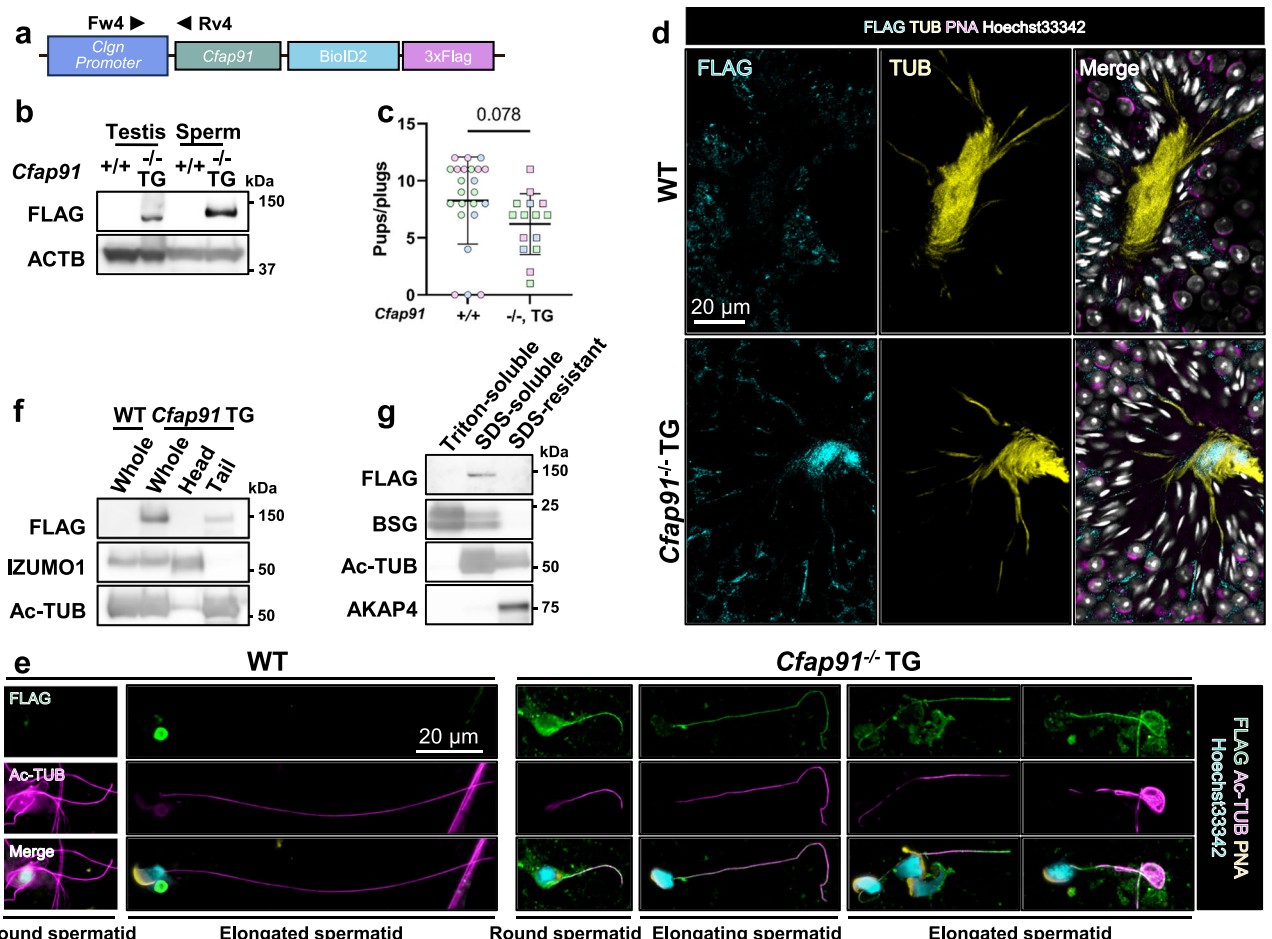

**Fig. 4 | CFAP91 is localized in sperm tails. a** A schematic diagram of the *Cfap91-BioID2-3×FLAG* transgene. Mice carrying this transgene are referred to as *Cfap91* TG mice. **b** Immunoblotting of CFAP91-BioID2-3×FLAG with an anti-FLAG antibody using testes or cauda epididymal spermatozoa of WT and *Cfap91⁻/⁻* TG males. **c** Fertility tests on *Cfap91⁺/⁺* and *Cfap91⁻/⁻* TG males. Data from each male mouse was individually color-coded. n = 23 plugs examined over 3 males for WT mice, n = 15 plugs examined over 3 males for *Cfap91⁻/⁻* TG mice. Data were presented as mean ± SD. An unpaired two-tailed t-test was performed for statistical analysis. **d** Immunohistochemistry on the sections of seminiferous tubules with an anti-FLAG antibody in WT and *Cfap91⁻/⁻* TG males. **e** Immunocytochemistry on spermatids obtained from WT and *Cfap91⁻/⁻* TG males with an anti-FLAG antibody. **f** Immunoblot analyses on sperm head-tail separated lysates. IZUMO1 and acetylated tubulin (Ac-TUB) served as controls of the head and tail fractions, respectively. **g** Immunoblot analyses on fractionated sperm lysates. BASIGIN (BSG), Ac-TUB, and AKAP4 served as controls for Triton-soluble, SDS-soluble, and SDS-resistant fractions, respectively.

elongation[52], these BBS proteins were only found in CFAP91 immunoprecipitates (Supplementary Fig. 5d). Furthermore, in *Cfap91⁺/⁺* testicular cells, BBS2 was present in the cytoplasm and along the sperm tails. Meanwhile, in *Cfap91⁻/⁻* testicular cells, the signal of BBS2 was not distributed evenly and formed foci in the cytoplasm (Supplementary Fig. 5e). These results suggest that CFAP91 may exhibit its effect on flagellum formation with BBS proteins.

### Proximity labeling in mature spermatozoa was able to detect EFCAB5 as a sperm-specific RS3 protein

Exploring protein bindings for sperm axonemal proteins in mature spermatozoa has been challenging, as typical lysis buffers that can solubilize axonemes may alter protein-protein interactions by changing the conformation of proteins[53]. To explore the interacting proteins of CFAP91, we applied proximity labeling in mature spermatozoa. For proximity labeling, we tagged BioID2 to CFAP91, which biotinylates the proteins surrounding CFAP91 in a radius of 10 nm in the presence of biotin[43]. Mature spermatozoa from *Cfap91⁻/⁻* TG cauda epididymis were cultured in a medium with biotin for 16 h to let BioID2 biotinylate its proximate proteins, then spermatozoa were lysed in a 0.4% SDS sample buffer. Subsequently, the biotinylated proteins were pulled down by streptavidin (SA) and analyzed by MS (Fig. 6a). To follow up, immunoblot analyses showed that CFAP91-BioID2-3×FLAG was pulled down by SA (Fig. 6b and Supplementary Data 3), indicating that self-biotinylation occurred. Twenty-four proteins were exclusively detected in the pulldown product of *Cfap91⁻/⁻* TG mice; among them, CFAP91 and EFCAB5 were identified with significantly more spectra than all proteins detected (Fig. 6c). We confirmed that EFCAB5 was pulled down by SA in mature spermatozoa of *Cfap91⁻/⁻* TG mice by immunoblotting (Fig. 6b). In addition, proximity labeling identified not only two RS3 proteins, CFAP251 and AKAP14, but also an RS1/2 protein, RGS22 (Fig. 6c, d), which was predicted to interact with EFCAB5 in the RS2-RS3 bridge of bovine spermatozoa[35] (Fig. 1a, b).

*Efcab5* is predominantly expressed in mouse testes (Fig. 6e). When we compared the expressions of *Efcab5*, *Cfap91*, *Cfap251* and *Lrrc23* in the testis with other abundantly expressed tissues (lung and brain), we found that *Efcab5* showed significantly higher expression ratio compared to other genes (Supplementary Fig. 6a). EFCAB5 is conserved among all mammals, 47 out of 58 Chordata, but it is missing in all of the Arthropoda included in TreeFam database[27] (Supplementary Fig. 6b). Moreover, EFCAB5 is not conserved in *C. reinhardtii* or *T. thermophila*. We confirmed that EFCAB5 exists in human spermatozoa by immunoblotting (Fig. 6f). Furthermore, immunoblot analyses revealed that EFCAB5 was absent in *Cfap91* KO spermatozoa (Fig. 6g). When sperm

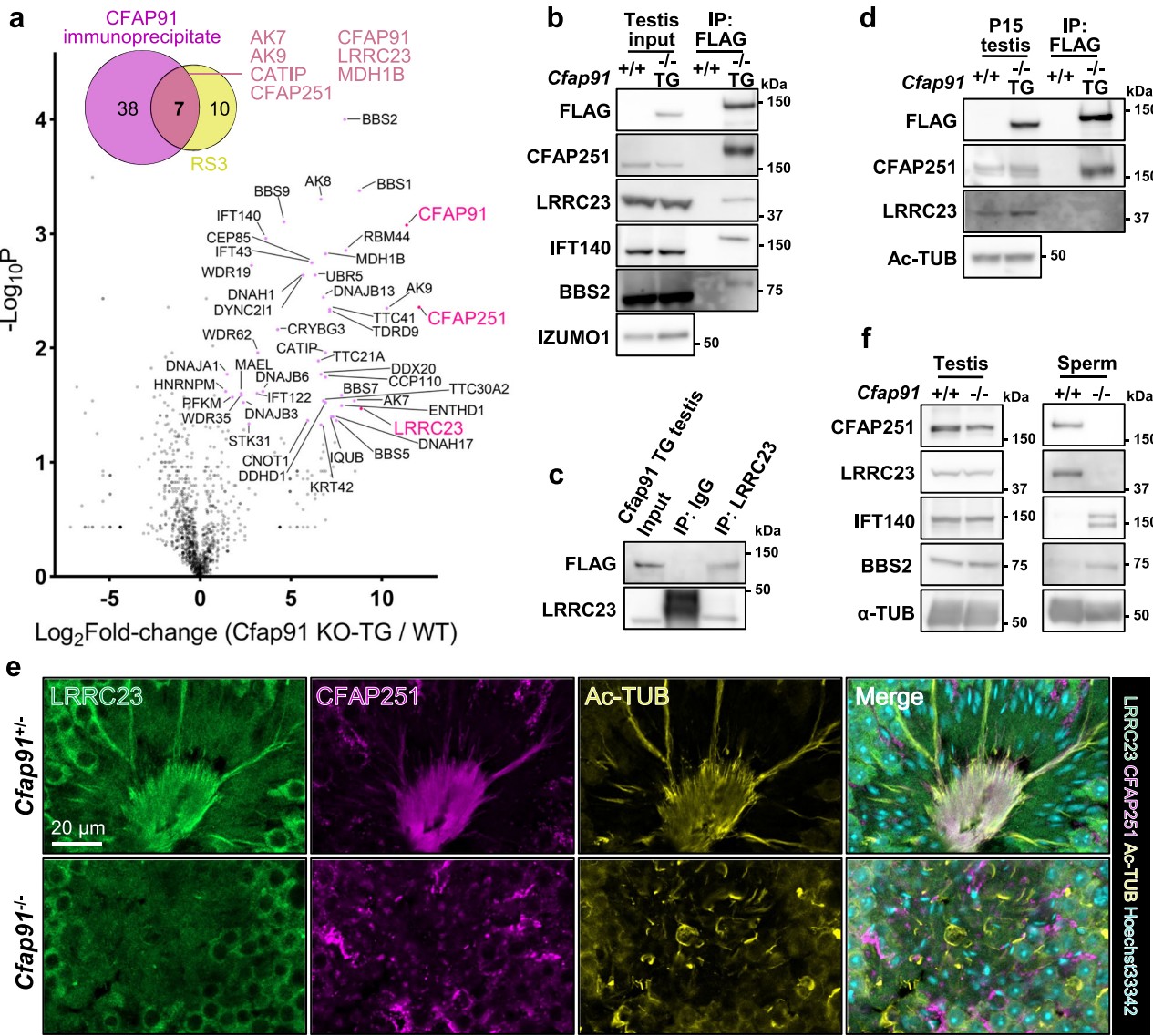

**Fig. 5 | CFAP91 immunoprecipitates with multiple RS3 proteins. a** A volcano plot of the results from IP-MS studies using an anti-FLAG antibody. When comparing *Cfap91^-/-* TG males to WT males, proteins with fold change >2 and *P* < 0.05 are considered as significantly upregulated, and dots are color-coded in magenta. An unpaired two-tailed t-test was performed for statistical analysis. **b** Immunoblotting was performed after IP with an anti-FLAG antibody. Signals of CFAP251, LRRC23, IFT140, and BBS2 were found in *Cfap91^-/-* TG but not WT. IZUMO1 served as a loading control of inputs. **c** Immunoblotting was performed after IP with an anti-LRRC23 antibody or rat IgG, on *Cfap91^-/-* TG testicular lysate. A band of FLAG was found in the IP product using the anti-LRRC23 antibody. **d** Immunoblotting was performed after IP using P15 testes from WT and *Cfap91^-/-* TG males. A band of CFAP251 was found in *Cfap91^-/-* TG but not WT, while bands of LRRC23 were not found in either *Cfap91^-/-* TG or WT. Acetylated tubulin (Ac-TUB) served as a loading control for input lysates. **e** Immunohistochemistry of *Cfap91^+/-* and *Cfap91^-/-* testicular sections. LRRC23 and CFAP251 were co-localized with Ac-TUB, while this co-localization was not found in Cfap91^-/- testicular sections. **f** Immunoblot analyses of testes and cauda epididymal spermatozoa from WT and *Cfap91^-/-* males. Alpha-tubulin (α-TUB) served as a loading control.

fractionation analyses were performed on WT mouse spermatozoa, EFCAB5, CFAP251, and LRRC23 all showed signals in the axoneme compartment like CFAP91 (Fig. 6h). These results indicate that EFCAB5 is a sperm-specific RS3 protein in mice.

### EFCAB5 is vital for sperm motility

According to a previous human male infertility study, *EFCAB5* expression is significantly downregulated in the testes of ejaculatory azoospermia patients[54], suggesting that EFCAB5 may play a role in human spermatogenesis. To verify its role in male fertility, we generated *Efcab5^-/-* mice. We electroporated Cas9 and two gRNAs that targeted the genomic region of *Efcab5* into 64 zygotes obtained from mating (B6D2F1 x B6D2F1) (Fig. 7a). We transferred 56 two-cell embryos into the oviduct of two pseudopregnant females and obtained 13 pups, 2 of which possess a large deletion in *Efcab5*. By

subsequent mating of F0 mice, *Efcab5^-/-* mice were obtained, with a large deletion in the genomic region of *Efcab5* confirmed (Supplementary Fig. 7a and b). It was observed that *Efcab5^-/-* males showed a slightly decreased fertility in mating test (Fig. 7b). No apparent anomalies were confirmed in the morphology of male reproductive organs (Supplementary Fig. 7c–g) or the mature spermatozoa in *Efcab5^-/-* males (Fig. 7c). In contrast to *Cfap91^-/-* males, CFAP251 and LRRC23 were present in the testes and spermatozoa of *Efcab5^-/-* males (Fig. 7d).

When we analyzed the motility of *Efcab5* KO spermatozoa (Supplementary Movie 1), we found a significant decrease in the percentages of both motile and progressive spermatozoa after 10 min and 120 min of incubation in capacitation medium (Fig. 7e). Moreover, curvilinear velocity (VCL), straight line velocity (VSL), and average path velocity (VAP) were decreased after 10 min of incubation; VSL and VAP

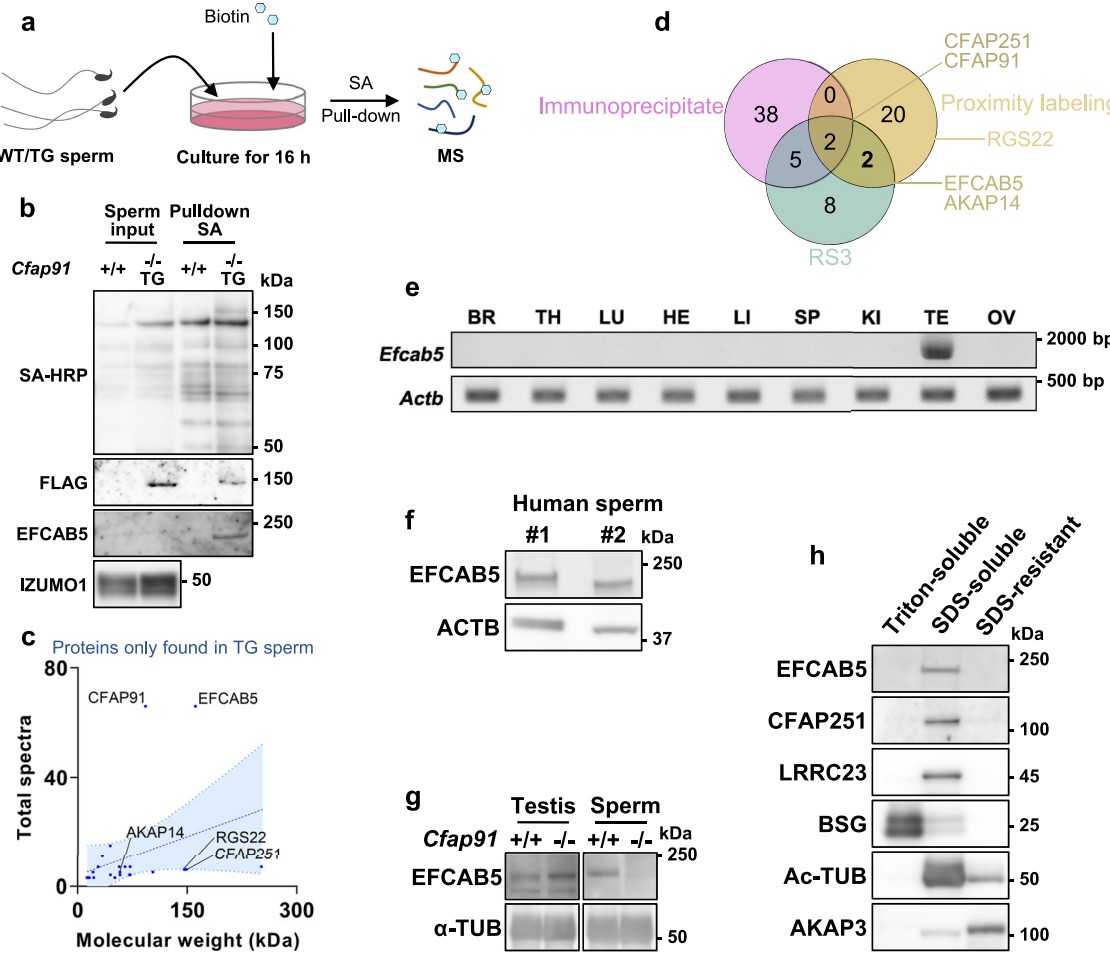

**Fig. 6 | Identification of EFCAB5 as a sperm axoneme-specific RS3 protein. a** A schematic drawing of the application of BioID2 in mature spermatozoa. Spermatozoa from WT and *Cfap91⁻/⁻* TG males were incubated in a medium supplemented with biotin for 16 h. Collected spermatozoa were lysed, and biotinylated proteins were pulled down by streptavidin (SA). **b** Immunoblotting was performed after the pull-down of biotinylated proteins in WT and *Cfap91⁻/⁻* TG males. EFCAB5 was not detected in the input, likely due to low protein abundance and/or low solubilization efficiency (0.4% SDS compared to 1% SDS or 6 M urea in other experiments). IZUMO1 served as a loading control for inputs. **c** Molecular weight and total spectra of twenty-four proteins only identified in *Cfap91* KO TG spermatozoa. CFAP91 and EFCAB5 were identified with the most peptides among all twenty-four proteins. A linear regression line with 95% confidence bands is shown. Slope = 0.06 and Y-intercept = 5.38. **d** Venn diagram of CFAP91 immunoprecipitates, proximity labeling-identified proteins, and mammalian sperm RS3 proteins. **e** RT-PCR of *Efcab5* utilizing cDNA from multiple mouse organs. *Actb* was used as a loading control. **f** Immunoblot analyses of human spermatozoa. **g** immunoblotting using testes and cauda epididymal spermatozoa from WT and *Cfap91⁻/⁻* males with an anti-EFCAB5 antibody. Alpha-tubulin (α-TUB) served as a loading control. **h** Fractionation using WT spermatozoa. Signals of EFCAB5, LRRC23, and CFAP251 were found in the SDS-soluble fraction.

were decreased after 120 min of incubation compared to WT spermatozoa (Fig. 7f). To test if the sperm motility pattern was also defective in *Efcab5⁻/⁻* males, we analyzed maximal bending angles of flagella relative to sperm heads (α-angle), which is used to quantitatively analyze hyperactivation[55,56]. Consistent with earlier reports, the α-angle of the control spermatozoa increased after 2 h of incubation in the capacitation medium, compared to 10 min of incubation[55,56]. However, this increase of α-angle was not seen in *Efcab5* KO spermatozoa (Fig. 7g), suggesting that the decreased fertility of *Efcab5⁻/⁻* males is caused by abnormal flagellar motility patterns.

As EFCAB5 localizes in the sperm-specific RS2-RS3 bridge, we propose that EFCAB5 may enable sperm-specific regulation of flagellar motility in this structure. In bovine spermatozoa, the RS2-RS3 bridge structure is formed by EFCAB5 and RGS22. It has been suggested that binding of EFCAB5 to RGS22 in RS2 may induce an intensive conformational change of RGS22[35]. We then analyzed the amount and localization of RGS22 in *Efcab5⁻/⁻* testes, but no overt abnormalities were found (Fig. 7d and Supplementary Fig. 8a). EFCAB proteins possessed one or multiple EF-hand calcium-binding domains, and we found that EFCAB5 was able to bind with Ca²⁺ in this domain according

to AlphaFold predictions[57] (Supplementary Fig. 8b). Moreover, Ca²⁺ influx occurs during capacitation to regulate downstream functions for hyperactivation[58,59]. To test if Ca²⁺ concentration can alter the EFCAB5-RGS22 binding, we expressed EFCAB5 and RGS22 in HEK293T cells and performed immunoblotting following IP in the presence and absence of Ca²⁺. As a result, RGS22 immunoprecipitated with EFCAB5 despite the concentration of Ca²⁺ (Supplementary Fig. 8c), indicating that EFCAB5 can bind to RGS22 independently of Ca²⁺.

## Discussion

Previously, human male patients lacking functional *CFAP91* have been reported as infertile[26]. However, the role of *Cfap91* in spermatogenesis has not been fully discovered. In this study, we employed the CRISPR/Cas9 system to generate *Cfap91⁻/⁻* mice and subsequently demonstrated that *Cfap91* is also essential for murine male fertility (Fig. 1g). *Cfap91⁻/⁻* mice exhibited abnormal sperm flagellum elongation (Fig. 1h, i), which resulted in short and immotile tails (Fig. 2a, b, d). Ultrastructural analyses on testicular and cauda epididymal spermatozoa showed that the integrity of sperm axonemes in *Cfap91⁻/⁻* males was damaged (Fig. 3a–f). As *Cfap91* KO spermatozoa showed abnormal

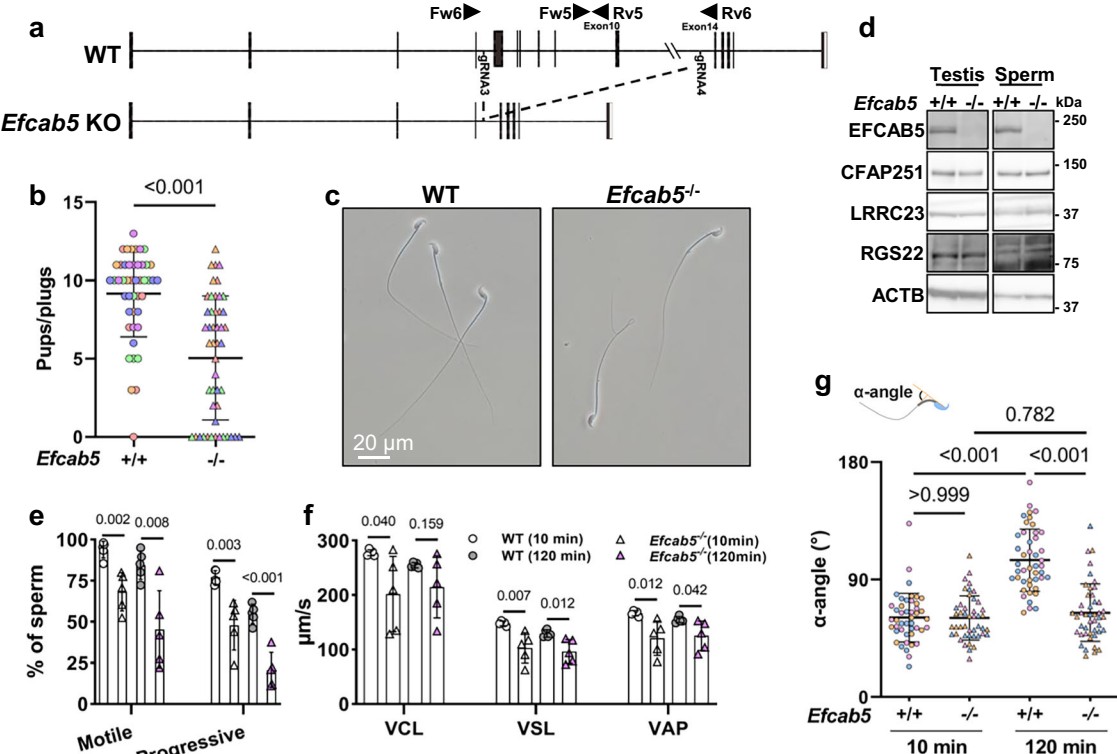

**Fig. 7 | EFCAB5 is vital for sperm motility. a** A schematic drawing of the KO strategy of *Efcab5*. **b** Fertility tests of *Efcab5*[+/+] and *Efcab5*[-/-] males. Data from each male mouse was individually color-coded. n = 46 plugs examined over 5 males for WT mice, n = 48 plugs examined over 5 males for *Efcab5*[-/-] mice. Data were presented as mean ± SD. An unpaired two-tailed t-test was performed for statistical analysis. *P* = 8.5E-8. **c** Phase contrast images of cauda epididymal spermatozoa from WT and *Efcab5*[-/-] males. **d** Immunoblotting of testes and cauda epididymal spermatozoa from WT and *Efcab5*[-/-] males, with anti-EFCAB5, anti-CFAP251, anti-LRRC23, and anti-RGS22 antibodies. ACTB served as a loading control. **e** Ratio of motile and progressive spermatozoa in WT and *Efcab5*[-/-] males after 10 min and 120 min of incubation in a capacitation medium. n = 5 males for each genotype. Data were presented as mean ± SD. An unpaired two-tailed t-test was performed for statistical analysis. *P* = 2.5E-04 for progressive motility after 120 min of incubation. **f** VCL, VSL, and VAP of cauda epididymal spermatozoa of WT and *Efcab5*[-/-] males after 10 min and 120 min of incubation in a capacitation medium. n = 5 males for each genotype. Data were presented as mean ± SD. An unpaired two-tailed t-test was performed for statistical analysis. **g** α-angle of cauda epididymal spermatozoa in WT and *Efcab5*[-/-] males after 10 min and 120 min of incubation in a capacitation medium. Data from each male mouse was individually color-coded. n = 45 spermatozoa examined over 3 males for each genotype and time point. Data were presented as mean ± SD. One-way ANOVA and Tukey's multiple comparisons test with adjustment were used for statistical analysis. *P* = 7.0E-14 for WT mice between 10 min and 120 min of incubation and *P* = 7.1E-14 for 120 min of incubation between WT and *Efcab5*[-/-] mice.

flagellum elongation, we could not identify if CFAP91 plays a role in regulating sperm flagellar motility. However, inactivation of *Cfap91* orthologs in *C. reinhardtii*, *Trypanosoma brucei*, and *T. thermophila* resulted in decreased mobility of flagella[26,30,60]. These earlier studies suggest that CFAP91 may also function in murine sperm motility regulation.

Imaging of manchette is usually performed by ICC or IHC using an anti-tubulin antibody[61]; yet, in this study, we generated *EGFP-Tuba3a* TG mice and demonstrated its usage in *Cfap91*[-/-] mice. As expected, *EGFP-Tuba3a* TG mice showed fluorescence in the manchette, and we were able to clarify that the abnormal sperm head shape was associated with abnormal manchette in *Cfap91*[-/-] males (Fig. 2h). Several KOs of sperm axonemal proteins that do not localize in the manchette exhibited defective manchette-mediated head morphogenesis[12,62], which suggests that CFAP91 may not be directly involved in the manchette elongation as well. Rather, the abnormal manchette formation may be a secondary effect of impaired flagellum formation. Intriguingly, EGFP-TUBA3A is not localized in sperm flagella (Fig. 2g), even though TUBA3A is considered a component of the sperm doublet microtubules[5]. This may be because N-terminally fused EGFP inhibits the uptake of TUBA3A into the doublet microtubules. Overall, the *EGFP-Tuba3a* TG mouse is a powerful tool for understanding manchette-mediated sperm head morphogenesis.

To understand the molecular function of CFAP91, we generated *Cfap91* TG mice. After performing IP-MS, we demonstrated that 45 proteins were significantly upregulated in CFAP91 immunoprecipitates (Fig. 5a). Of these 45 proteins, we especially focused on CFAP251 and LRRC23. Previously, CFAP251 orthologs were found to interact with CFAP91 orthologs in *C. reinhardtii* and *T. thermophila*[28,29]. This interaction was found to be conserved in mouse ependymal cilia and human respiratory cilia[31,32]. In these four species, CFAP251 was hypothesized to be localized at the base of RS3, suggesting that the localization of CFAP251 in the RS3 stalk is conserved from protists to mammals. CFAP251 was downregulated in the spermatozoa of human patients carrying a pathogenic variant in *CFAP91*[26]. Furthermore, human male infertile patients carrying *CFAP251*-deficient alleles were found to have shorter sperm tails[63], indicating that CFAP251 is essential for sperm flagellum formation. Hence, depletion of CFAP91 disabled the incorporation of CFAP251 into the sperm flagella (Fig. 5e), which may be the essential reason that *Cfap91*[-/-] males produced shorter sperm tails. In contrast, LRRC23 truncation has been shown to cause the loss of the RS3 head in mouse spermatozoa without overt abnormalities in flagellum elongation[11]. Recently, structural analyses on bovine spermatozoa reported that LRRC23 localizes in the head of RS3 and was supported by the C-terminus of CFAP91 that localizes along the RS3 neck[35] (Fig. 1b). We confirmed that LRRC23 was pulled down by the C-terminal half of CFAP91 (Supplementary Fig. 8d).

However, the physical interaction between CFAP91 to LRRC23 was not found in bovine sperm RS3[35], suggesting that the interaction between CFAP91 and LRRC23 is indirect.

Interestingly, in *C. reinhardtii*, a shorter RS3 can be formed only by orthologs of CFAP61/91/251[31], still they are enough for flagellum elongation. Furthermore, in mammalian RS3, only the absence of CFAP61/91/251[26,62–64] impairs sperm tail length, but not other known RS3 members (AK9[65], AKAP14[66], LRRC23[11,67], LRRD1[68], PKA[69,70], STYXL1[71]; it is not clear for *Ak7[-/-]* males since the study was conducted at 42-day-old mice, and full-size sperm tails may be formed in adult mice[72]), suggesting that the basic unit sufficient for flagellum elongation may be CFAP61/91/251 in mouse spermatozoa. In the P15 testes before the appearance of sperm tails, we were able to immunoprecipitate CFAP251 but not LRRC23 with CFAP91 (Fig. 5d). This conveys that CFAP251-CFAP91 may be assembled as an integral part of the axoneme elongation. In contrast, LRRC23, which tunes motility, may later be assembled on the C-terminus region of CFAP91 that extends along the RS3 neck. It should be noted that we were also able to detect CFAP61 in IP-MS samples with high fold change, although it was not statistically significant (Supplementary Data 1; $P = 0.13$).

Based on our results that indicate CFAP91 localization to the mouse sperm RS3, we performed proximity labeling experiments using mature spermatozoa, which led us to identify EFCAB5 with high peptide numbers (Fig. 6c). When we were preparing this manuscript, an atomic model of bovine sperm RS3 was published[35], which also indicates the localization of EFCAB5 in the RS3 (Fig. 1b). We also detected EFCAB5 with IP-MS using *Cfap91* TG mice, but it was not statistically significant compared to control WT mice (Supplementary Data 1; $P = 0.08$). The reason why EFCAB5 was not significantly enriched in CFAP91 immunoprecipitates (Fig. 5a) could be multifactorial. The most likely cause may be the existence of numerous types of spermatogenic cells in testicular IP-MS analyses. The highly organized axoneme structures are formed in stages, and EFCAB5 may attach to the basic structure of the RS3 in later stages, in which the flagella are highly compacted and not soluble in the IP lysis buffer. It is possible that EFCAB5 is evolutionarily appended to the RS3 basic structure when RS3 evolved from its ancestors, consistent with EFCAB5 being a critical component for sperm-specialized movement. Our analyses using HEK293T cells confirmed that the interaction of EFCAB5 and RGS22, an RS2 protein, is independent of $Ca^{2+}$ (Supplementary Fig. 8c). The constant connection of RS2 and RS3 through RGS22/EFCAB5 may be important for the regulation of sperm-specialized movement.

In summary, this study suggests that CFAP91 localizes to sperm RS3 and is involved in sperm tail biogenesis by recruiting CFAP251 into the assembling axoneme. Moreover, we report an *EGFP-Tuba3a* TG mouse line that can be used for the live imaging of the manchette. This study also demonstrates proximity labeling in mature spermatozoa, which has the advantage of detecting adjacent structural proteins in axonemes that are difficult to dissolve in mild lysis buffers. Our proximity labeling analysis indicates that EFCAB5 is a sperm axoneme-specific RS3 protein that is vital for sperm motility. This study widens the understanding of etiology and pathobiology of human male infertility.

## Methods

### Maintenance and generation of mice
All mouse experiments were approved by the Animal Care and Use Committee at the Research Institute for Microbial Diseases, Osaka University (Approval numbers: #Biken-AP-H30-01 and #Biken-AP-R03-01). BDF1 and ICR mice were purchased from Japan SLC (Shizuoka, Japan) and CLEA Japan (Tokyo, Japan). Mice in the 129×1/SvJmsSlc background were purchased from Japan SLC. All mice were housed in a specific-pathogen-free animal facility and were able to access unlimited food and water voluntarily. The animal room was maintained on a 12-h light/12-h dark cycle. Male mouse experiments were conducted when they reached 8 weeks of age.

To generate KO mice with the CRISPR/Cas9 system, pairs of gRNAs were designed to target the genomic region of *Cfap91* and *Efcab5*. Each gRNA was incubated with Cas9 (Thermo Fisher Scientific, Waltham, MA) and trans-activating crRNA (tracrRNA) (Sigma-Aldrich, St Louis, MO) to induce the formation of Cas9-gRNA ribonucleoprotein (RNP). RNPs are subsequently electroporated into mouse zygotes, which are obtained from the natural mating of BDF1 males with superovulated BDF1 female mice. For the generation of *EGFP-Tuba3a* TG mice, *EGFP-Tuba3a* cDNA was cloned into an expression vector containing a *Clgn* promoter (Addgene #173686). For *Cfap91* TG, *Cfap91* cDNA was inserted into CAG-MCS-BioID2-3×FLAG plasmid[73] (Addgene #186812). Then, the CAG promoter was replaced by a *Clgn* promoter[74] (Addgene #173686). Both plasmids were linearized by ClaI and PacI (New England Biolabs, Essex, MA), and pronuclear injection of linearized plasmids was performed on mouse zygotes. In both cases of generating KO and TG mice, the zygotes were cultured to the two-cell stage in potassium-supplemented simplex optimized medium[75] (KSOM) after electroporation or injection and then transplanted into the ampulla of pseudopregnant ICR female mice. Sanger sequencing and genomic PCR of offspring obtained from pseudopregnant females were performed to check their genotype. Genomic PCR was performed with the following conditions for 40 cycles: 94 °C for 30 s, 65 °C for 30 s, 72 °C for 30 s (except for F3–R3, as the last step was extended to 120 s). The sequences of gRNAs and primers are listed in Supplementary Data 4.

### RT-PCR
Total RNA from various tissues and testicular RNA in postnatal testes were extracted by TRIzol (Thermo Fisher Scientific) and reversed transcribed by SuperScript III First-Strand Synthesis System (Thermo Fisher Scientific). Products from reverse transcription were used as templates for PCR. PCR cycles were set in the following conditions: 94 °C for 30 s, 60 °C for 30 s, and 72 °C for 60 s (For *Actb*, the last step was set to 30 s). The sequences of primers used for RT-PCR are listed in Supplementary Data 4.

### Fertility test
For the *Cfap91* KO and *Cfap91* TG mouse lines, the fertility (pups/plugs) of three males was tested. For the *Efcab5* KO mouse line, the fertility of five KO males was tested. Each male was housed with three WT females for 11 weeks, and the presence of vaginal plugs and the number of offspring were counted every weekday.

### Histological analysis
Testis and epididymis specimens were immersed in Bouin's Fixative (Polysciences, Inc., Warrington, PA) for 6 h at room temperature. After gradual dehydration, specimens were embedded in a paraffin block and sliced into sections with a thickness of 5 μm. Sections were dewaxed and treated with 1% periodic acid (Nacalai Tesque, Kyoto, Japan) for 10 min, Schiff's reagent (Wako, Osaka, Japan) for 20 min, and Mayer's Hematoxylin Solution (Wako) for 30 s, with a brief wash by tap water after each step. Sections were mounted with Entellan new (Sigma-Aldrich) and observed by an Olympus BX-53 microscope (Tokyo, Japan).

### Immunohistochemistry (IHC) and immunocytochemistry (ICC)
For IHC, tissues were immersed in 4% paraformaldehyde (PFA; Thermo Fisher Scientific) for 6 h at 4 °C, and were transferred to 15% and 30% sucrose for a day at 4 °C before the samples were embedded with Optimal Cutting Temperature (OCT) Compound (Sakura Finetek, Tokyo, Japan). Specimens were then sliced into 8 μm-thick sections and briefly washed with phosphate-buffered saline (PBS). Then, the specimens were treated with 0.1% Triton X-100 (Nacalai Tesque) in PBS and were blocked with 3% bovine serum albumin (BSA; Sigma-Aldrich) in PBS. Then, the specimens were incubated overnight at 4 °C with

primary antibodies that were diluted in 3% BSA in PBS (Supplementary Data 5). On the following day, the specimens were washed with PBS and incubated for 2 h with secondary antibodies that were diluted in 3% BSA in PBS. Finally, the specimens were washed with PBS, incubated with 0.05% of Hoechst 33342 (Thermo Fisher Scientific), and mounted with Epredia Immu-Mount (Thermo Fisher Scientific) prior to the observation under an Olympus BX-53 microscope or a Nikon C2 Eclipse Ti microscope (Nikon, Tokyo, Japan).

For ICC, spermatozoa and spermatids were spread onto slides and were air-dried before fixation with 4% PFA for 15 min at room temperature. The slides were washed with PBS, blocked, probed with antibodies, and mounted as described in IHC.

### Sperm parameter inspection
To measure sperm tail length, the ratio of abnormal sperm heads, and sperm count, we released mature spermatozoa from the cauda epididymis into PBS using fine forceps and scissors. Spermatozoa were then spread onto a slide, and abnormal sperm heads were inspected and photographed using an Olympus BX-53 microscope. Tail lengths were analyzed using Fiji[76]. Spermatozoa were further diluted in water for immobilization, and the number of spermatozoa was counted using a hemocytometer.

For sperm motility measurements, cauda epididymal spermatozoa were released into TYH capacitation medium[77] and incubated at 37 °C under 5% $CO_2$ in air. For *Cfap91*[-/-] males, we measured sperm motility after only 10 min of incubation, as spermatozoa lost motility completely. For *Efcab5*[-/-] males, spermatozoa were incubated for 10 min and 120 min, which represents before and after capacitation[78]. We analyzed sperm motility using the CEROS II sperm analysis system (software version 1.5; Hamilton Thorne Biosciences, Beverly, MA). For the analysis of the α-angle of spermatozoa, we captured videos of sperm movement with an Olympus BX-53 microscope equipped with a high-speed camera HAS-L1 (Ditect, Tokyo, Japan). The video was recorded at 200 frames per second. The α-angle of spermatozoa was analyzed using Fiji[76].

### Imaging of manchette using *EGFP-Tuba3a* TG male mice
Seminiferous tubules of *EGFP-Tuba3a* TG males were released by breaking the tunica albuginea. Seminiferous tubules were gently pipetted to remove the attachment of Leydig cells and were squeezed by fine forceps to release the spermatids into PBS. Spermatids were then stained with 0.1% Hoechst 33342 and imaged with an Olympus BX-53 microscope.

### Ultrastructural analysis
Ultrastructural analysis by TEM was performed as previously described[79]. In brief, anesthetized male mice underwent perfusion fixation using 4% PFA, after which the testes were further fixed in 4% PFA. The samples were fixed again with 1% glutaraldehyde. Further, the testes were immersed in a solution of 1% osmium tetroxide ($OsO_4$) and 0.5% potassium ferrocyanide. Following dehydration with ethanol, the specimens were incubated in propylene oxide (PO) before being placed in a mixture of PO and epoxy resin. This mixture was subsequently replaced with pure epoxy resin. Ultrathin sections were cut and stained, and the images were captured using a JEM-1400 Plus electron microscope (JEOL, Tokyo, Japan) operating at 80 kV, equipped with a Veleta 2 K × 2 K CCD camera (Olympus).

### Immunoblotting (IB)
IB was performed as previously described[80]. In the case that the lysis buffer was not specified, testicular proteins and sperm proteins were lysed by Triton lysis buffer [1% Triton X-100, 50 mM NaCl, 20 mM Tris-HCl, 1× protease inhibitor cocktail (Nacalai Tesque)] and urea lysis buffer (6 M urea, 2 M thiourea, 2% sodium deoxycholate), respectively. For human spermatozoa, samples were obtained from fertile donors with their consent. The experiments using human samples were approved by the Ethics Committee of the Research Institute for Microbial Diseases, Osaka University (#28-4-2). The uncropped and unprocessed scans of IB are provided in the Source Data file.

### Sperm head-tail separation
Cauda epididymal spermatozoa were collected in PBS and sonicated by Branson SLPe Digital Sonifier (Branson Ultrasonics, Brookfield, CT). Sonicated samples were gently transferred to the top of 90% Percoll (GE Healthcare, Chicago, IL) in PBS and centrifuged to separate sperm heads and tails. Finally, the separated heads and tails were resuspended in a sample buffer (60 mM Tris-HCl pH 6.8, 2% SDS, 10% glycerol, 0.025% Bromophenol Blue) and incubated at 95 °C for 5 min for protein extraction. A detailed protocol can be found in the previous report[3].

### Sperm fractionation
Sperm fractionation was performed as previously reported[52]. In brief, cauda epididymal spermatozoa were treated in the order of Triton lysis buffer, 1% SDS lysis buffer (1% SDS, 75mM NaCl, 24 mM EDTA, 1× protease inhibitor cocktail), and sample buffer, at 4 °C for 2 h, room temperature for 1 h, and 95 °C for 5 min, respectively. Samples were centrifuged, and the supernatant was taken and replaced with subsequent lysis buffer after the first and second protein extraction.

### Cell maintenance and transfection
HEK293T cells[81] were cultured in Dulbecco's Modified Eagle Medium (DMEM; Thermo Fisher Scientific) supplemented with 10% fetal bovine serum (FBS; Sigma-Aldrich) and 1% penicillin and streptomycin (Thermo Fisher Scientific) at 37 °C under 5% $CO_2$ in air and was sub-cultured every 3 or 4 days. Transfection of cells was carried out by the calcium phosphate-DNA precipitate method when cells reached 70–80% confluency. Cells were harvested 24 h post-transfection and were subjected to subsequent experiments.

### Immunoprecipitation (IP)
Magnetic beads-based IP was performed. Cell lysates were subjected to IP using the Dynabeads Co-Immunoprecipitation Kit (Thermo Fisher Scientific). IP was carried out under the recommended protocol of the manufacturer. The IP product was subjected to immunoblotting or MS analyses.

### Antibodies
Antibodies used in this study are indicated in Supplementary Data 5. Antibodies against BASIGIN[82], EGFP[83], IZUMO1[84], LRRC23[67] were generated in the previous studies. The antibody against EFCAB5 was generated in this study against the sequence 'CGSRRGSGTDQGQHRGSV'. The Anti-1D4[85] antibody was a gift from Dr. Martin M. Matzuk.

### Mass spectrometry
Proteins were analyzed by nanocapillary reversed-phase liquid chromatography-tandem mass spectrometry (LC-MS/MS). The analysis was performed using a C18 column (IonOpticks, Victoria, Australia) integrated into a nanoLC system (Bruker Daltonics, Billerica, MA), which was coupled to a timsTOF Pro mass spectrometer (Bruker Daltonics) and the CaptiveSpray nano-electrospray ion source (Bruker Daltonics). The raw data were processed using DataAnalysis software (Bruker Daltonics), and protein identification was carried out using MASCOT (Matrix Science, Tokyo, Japan) with the SwissProt database. Quantitative values and fold changes were subsequently calculated using Scaffold 5 (Proteome Software, Portland, OR).

### Proximity labeling of cauda epididymal spermatozoa
Cauda epididymal spermatozoa were collected into TYH medium supplemented with 250 μM biotin (Sigma-Aldrich) and incubated at 37 °C under 5% $CO_2$ in air for 16 h. Subsequently, spermatozoa were

collected and washed with PBS twice. Protein extraction from spermatozoa was performed according to earlier reports[43], with several modifications. SDS sample buffer [50 mM Tris (pH 7.4), 500 mM NaCl, 0.4% SDS, 1 mM dithiothreitol (DTT), and 1× protease inhibitor cocktail] was used as a lysis buffer. After incubating spermatozoa in the lysis buffer for 1 h at 4 °C, Triton X-100 was added to the lysis buffer till Triton X-100 reached 2% for the final concentration. Tubes were then intensely vortexed for 2 min. Then, 50 mM Tris (pH 7.4) was added to the tube until the final concentration of Triton X-100 reached 1%. Samples were then briefly vortexed and centrifuged at 15,000 × *g* for 20 min, and the supernatant was collected and incubated with Pierce Streptavidin Magnetic Beads (Thermo Fisher Scientific) overnight at 4 °C. After incubation, beads were washed with wash solutions three times for streptavidin pull down. The recipe for wash solutions can be found in the indicated paper[86]. Followed by the three washes, 2× SDS sample buffer was incubated with the beads at 95 °C for 5 min to elute the proteins attached to the beads. Eluates were then analyzed by immunoblotting and MS.

## Structural analysis

Cryo-EM density maps and fitted models were visualized by UCSF ChimeraX[87].

## Statistical analysis and reproducibility

All experiments were performed at least three times with similar results. Statistical analysis was performed using Graphpad Prism 9 (GraphPad Software, MA, USA). Data are all indicated as mean ± SD. An unpaired two-tailed t-test was performed for statistical analyses unless otherwise specified. The level of significance was set at $P < 0.05$, and the $P$ values were directly indicated in the figures.

## Reporting summary

Further information on research design is available in the Nature Portfolio Reporting Summary linked to this article.

## Data availability

The authors declare that the data that support the findings of this study are available from the corresponding author upon request. Proteomic data is available with accession codes JPST003879 and PXD065243. Cryo-EM density maps and fitted model were retrieved from previous studies with accession codes EMDB-35888 [https://www.ebi.ac.uk/emdb/EMD-35888][31], EMDB-50664 [https://www.ebi.ac.uk/pdbe/entry/emdb/EMD-50664] and 9FQR[35]. Source data are provided with this paper.

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

## Acknowledgements

We thank Keiko Murata for sequence analysis, Hiroko Omori for ultrastructural analysis, Akinori Ninomiya and Hiroko Kato for mass spectrometry analysis (Core Instrumentation Facility, Research Institute for Microbial Diseases, Osaka University). We also thank Natsuki Furuta and Kaito Yamamoto for their technical assistance, and Ferheen Abbasi and Joanna J. Luna for critical reading of this manuscript. We appreciate the experimental discussion with the members of the Department of Experimental Genome Research. This study was supported by the Japan Society for the Promotion of Science (JSPS) KAKENHI grants (JP23K05831 to K.S., JP22H03214, JP23K18328, JP25K02773 to H.M., and JP19H05750, JP21H04753, JP21H05033, JP23K20043 to M.I.); Japan Agency for Medical Research and Development (AMED) grant (JP23jf0126001 to M.I.); a Takeda Science Foundation grant to K.S., H.M., and M.I.; JST FOREST (JPMJFR211F to H.M.); the Eunice Kennedy Shriver National Institute of Child Health and Human Development (P01HD087157 and R01HD088412 to M.I.); and the Bill & Melinda Gates Foundation (Grand Challenges Explorations grant INV-001902 to M.I.). Molecular graphics and analyses performed with UCSF ChimeraX, developed by the Resource for Biocomputing, Visualization, and Informatics at the University of California, San Francisco, with support from National Institutes of Health R01-GM129325 and the Office of Cyber Infrastructure and Computational Biology, National Institute of Allergy and Infectious Diseases.

## Author contributions

Conceptualization, H.W., K.S., H.K., S.O., N.Y., M.I., and H.M.; Formal analysis, H.W., K.S., A.P., M.K., H.K., and S.O.; Investigation, H.W., K.S., A.P., Y.O., M.K., and H.M.; Supervision, N.Y., M.I., and H.M.; Funding acquisition, K.S., M.I., and H.M.; Manuscript preparation, H.W., K.S., M.I., and H.M.

## Competing interests

The authors declare no competing interests.
