## [Peer Review file · Nature Communications]

Proximity labeling of axonemal protein CFAP91 identifies EFCAB5 that regulates sperm motility

Corresponding Author: Professor Haruhiko Miyata

Version 0:

Reviewer comments:

Reviewer #1

(Remarks to the Author)

Wang and colleagues present a comprehensive analysis of mouse CFAP91, an axonemal protein implicated in human male infertility. They generated a Cfap91 knockout mice and showed that male mice are infertile due to failed spermiogenesis. They were able to rescue this phenotype with a Cfap91 transgene that encodes CFAP91-BioID2-3xFLAG. Using the tagged version of CFAP91, the authors perform two different mass spectrometry experiments: immunoprecipitation-mass spectrometry (IP-MS) from testis lysate and proximity labeling in mature spermatozoa. Collectively, these proteomic studies identified a number of potential CFAP91 interactors including CFAP251, LRRC23, and EFCAB5. Following up on these observations, the authors generated Efcab5 knockout male mice and demonstrated that they were subfertile, potentially due to abnormal flagellar motility patterns.

Overall, this is a valuable study built on convincing evidence that deserves publication. However, as described below, some improvements could be made to how the paper is organized to incorporate recently published structural data on CFAP91 and its interactions in radial spoke 3.

Major comments

1. The authors should consider reorganizing their paper to describe the recent structural information elucidating the composition of sperm radial spoke 3 (Leung et al, Nature, 2025) in the introduction, and then use this information to interpret their findings. The advantage of this approach is that it would allow better annotation of their mass spectrometry results (e.g. explaining why they identify AK7, AK9, CATIP, CFAP251, LRCC23, and MDH1B by IP-MS and why they identify EFCAB5, AKAP14, and RGS22 by BioID). I believe this reorganization would not remove novelty from the paper but provide robust rationale for the subsequent experiments.
2. Even in the absence of describing the recent findings of Leung et al in the introduction, the authors need to describe more accurately what is known about the interactions of CFAP91 from structural information in other species. For example, Walton et al, Nature, 2023 describe the interactions of FAP91 in *C. reinhardtii* flagella and CFAP91 in human respiratory cilia axonemes, including the interaction with CFAP251.
3. The authors write “CFAP91 also showed interactions with RSPH9, which has been discovered to be localized at the RS3 head”. This statement is unfortunately based on inaccurate modeling of low-resolution cryo-ET data of mouse ependymal cilia. Leung and colleagues have demonstrated that the head of RS3 does not contain RSPH9.
4. The work of Leung and colleagues did not report a direct interaction between CFAP91 and LRRC23 in the bovine sperm axoneme (although both exist in the same radial spoke). This discrepancy between results should be discussed.
5. Related to Comments 1-4, the Abstract needs to be rewritten to better represent the current literature.
6. The results of the IP experiments are overinterpreted as showing an “interaction” with CFAP91. However, because some of these interactions are likely indirect, a more cautious approach to interpreting these results should be adopted. For example, the authors could replace “interact” with “co-precipitate”.

7. P4. L74. The authors state that they have “revealed the function of CFAP91”, which in my opinion does not accurately reflect the conclusions of the work. They have identified potential binding partners of CFAP91 and identified that it is critical for sperm tail assembly, but these points are distinct from identifying the “function” of CFAP91.
8. Videos must be provided showing the movement of spermatozoa from the *Efcab5*^{-/-} and control mice.
9. The proteomics data should be deposited in public repositories (e.g. PRIDE).

Minor comments

1. “Consistently” and “Moreover” are overused in the text, sometimes in consecutive sentences
2. Do the authors have RT-PCR data for the choroid plexus and oviduct? The Human Protein Atlas indicates that CFAP91 expression is particularly high in these tissues.
3. Given the relatively small number of proteins identified by IP-MS, the GO analysis of the CFAP91 interactome (Fig. S5b) could be replaced by manual annotation of the proteins, e.g. IFT-related proteins, radial spoke proteins, other axonemal.
4. *C. reinhardtii* is consistently misspelt as *C. reinhartii*
5. This sentence starting on page 3, line 41 is circular. “indicating that the sperm axoneme is different from the ciliary axoneme” can be removed as the point is already made by the first half of the sentence.
6. P4. L70. “whether CFAP91 shows the same localization in sperm flagella remained unknown” should be updated in light of recent structural studies.
7. P9. L200. Immunoblot is not verifying the interaction CFAP91 with CFAP251, LRRC23, and IFT140, but simply detecting the presence of these proteins in the immunoprecipitation by a different method
8. P10.L225. *hault* -> *halt*
9. P14.L337. The authors state that EFCAB5 was not found in the IP-MS study, but Table S1 suggests otherwise. It was detected with a Log2fold-change of 4.12 and p-value of 0.082.

Reviewer #2

(Remarks to the Author)

In this study, Wang et al. investigate the functional role of CFAP91 in spermatogenesis. They demonstrate that CFAP91 interacts with multiple RS3 proteins within the sperm axoneme. Genetic ablation of *Cfap91* in mice results in spermatozoa with shortened tails and abnormal head morphology. Through a CFAP91-targeted proximity labeling strategy, the authors identify a sperm-specific RS3 protein, EFCAB5, and show that its depletion impairs sperm motility. While the experimental observations are technically sound, the molecular mechanisms remain inadequately explored. Furthermore, the study does not appear to represent a substantial mechanistic advancement in this field.

Major Concerns

- 1) The authors fail to fully elucidate how CFAP91 participates in sperm flagellar biogenesis. Precise spatiotemporal localization of CFAP91 during spermiogenesis may be helpful to distinguish between direct versus indirect roles in axoneme assembly. Although the CFAP91 interactome was characterized in mouse testis, functional validation of its binding partners in flagellar formation remains lacking. Notably, while CFAP91 depletion disrupts CFAP251/LRRC23 localization in flagella, *Lrrc23* knockout mice exhibit normal sperm morphology (PMID: 34585727), suggesting LRRC23 may act downstream of CFAP91. The observed absence of radial spoke proteins in *Cfap91* KO sperm may alternatively result from axonemal structure defects. Thus, critical binding partners directly mediating CFAP91's role in flagellar biogenesis should be experimentally identified.
- 2) Multiple BBSome components (BBS1, BBS2, BBS4, BBS7) were identified in the CFAP91 interactome. Given established roles of BBSome proteins in flagellar assembly (PMID: 18032602, 15173597, 23572516, 15539463), the authors need to test whether CFAP91 exerts its effects through BBSome-dependent mechanisms.
- 3) The sub-Mendelian ratio of *Cfap91*^{-/-} mice in the B6D2 background and potential hydrocephalus phenotypes raise concerns about CFAP91's broader roles in ciliogenesis. The authors should clarify whether CFAP91 depletion affects primary cilia formation in the B6D2 background and determine if its regulatory mechanisms are conserved between ciliogenesis and sperm flagellar biogenesis. Cell-based ciliogenesis models could provide mechanistic insights beyond current *in vivo* approaches.
- 4) While EFCAB5 is identified as a novel RS3 component, and its knockout reduces sperm motility, the molecular mechanisms of EFCAB5 in sperm movement remains unaddressed. The authors propose that “EFCAB5 may regulate sperm-specific motility patterns by permitting the conformational change of RGS22 which links RS3 to RS2.” Whether EFCAB5 depletion affects the RGS22 or other RS protein localization?
- 5) Emerging structural model from bovine sperm flagella (PMID: 39743588) propose domain-specific interactions of CFAP91: its N-terminal region potentially bridges the DMT inner junction, RS2, and N-DRC (implicating roles in axoneme assembly), while the C-terminal domain may recruit motility regulators like EFCAB5. Experimental validation of these predicted interaction domains is essential to unify the study's two major findings.

Minor Concerns

- 1) Figures 1a and 6i: Incorporate recent structural models from PMID: 39743588 to enhance schematic accuracy.
- 2) The authors state that “abnormal morphology of Cfap91 KO sperm heads was likely caused by abnormally elongated manchettes.” However, in the discussion part, it is discussed as “this abnormal manchette elongation may be a secondary effect due to impaired flagellum formation”. This contradiction requires resolution.
- 3) Figure 2a shows apparently normal sperm heads in Cfap9^{-/-} mice, whereas Figure 2c reports near 100% of head abnormalities. Representative images should be re-examined.
- 4) Figure 5c, IgG in ^{-/-} TG mice may be a better negative control than ^{+/+} mice.
- 5) Figure 6b, EFCAB5 signal absence in “Sperm Input” lanes necessitates either longer exposure or alternative blot presentation.

Version 1:

Reviewer comments:

Reviewer #1

(Remarks to the Author)

In my opinion, the manuscript has undergone considerable improvement since its initial submission. I appreciate the efforts the authors have made to incorporate recent structural data into their analysis of the results. I recommend publication.

During reading the revised manuscript, I encountered a few sentences that the authors might want to correct/revise:

L19. It is incorrect to say that radial spokes are “attached to the axoneme” – they are part of it. They are attached to doublet microtubules.

L20. “despite the presence of additional structures on the sperm axoneme” is unnecessary and confusing in the abstract without more information. I would recommend deleting.

L39. It might be more accurate to write “motile cilia” than “the cilia”

L66. “the underlying mechanism of CFAP91 is not fully understood” – the authors should define what they mean by “mechanism”. Do they mean the mechanism by which CFAP91 regulates ciliary motility? Or the disease mechanism that leads to male infertility?

L77. The authors neglect to mention the contribution CFAP91 makes to the N-DRC.

Reviewer #2

(Remarks to the Author)

The authors have addressed my concerns.

Thank you for the opportunity to revise our manuscript entitled "Proximity Labeling of Axonemal Protein CFAP91 Identifies EFCAB5 that Regulates Sperm Motility". We sincerely appreciate the reviewers' constructive feedback and have revised our manuscript per these comments. We have provided point-by-point detailed responses (in black) to the reviewers' comments (in blue). The line numbers from the main manuscript with track changes (the file was uploaded in "Related Manuscript File") are indicated below.

Reviewer #1 (Remarks to the Author):

Wang and colleagues present a comprehensive analysis of mouse CFAP91, an axonemal protein implicated in human male infertility. They generated a Cfap91 knockout mice and showed that male mice are infertile due to failed spermiogenesis. They were able to rescue this phenotype with a Cfap91 transgene that encodes CFAP91-BioID2-3×FLAG. Using the tagged version of CFAP91, the authors perform two different mass spectrometry experiments: immunoprecipitation-mass spectrometry (IP-MS) from testis lysate and proximity labeling in mature spermatozoa. Collectively, these proteomic studies identified a number of potential CFAP91 interactors including CFAP251, LRRC23, and EFCAB5. Following up on these observations, the authors generated Efcab5 knockout male mice and demonstrated that they were subfertile, potentially due to abnormal flagellar motility patterns.

Overall, this is a valuable study built on convincing evidence that deserves publication. However, as described below, some improvements could be made to how the paper is organized to incorporate recently published structural data on CFAP91 and its interactions in radial spoke 3.

Thank you very much for your comments.

Major comments

1. The authors should consider reorganizing their paper to describe the recent structural information elucidating the composition of sperm radial spoke 3 (Leung et al, Nature, 2025) in the introduction, and then use this information to interpret their findings. The

advantage of this approach is that it would allow better annotation of their mass spectrometry results (e.g. explaining why they identify AK7, AK9, CATIP, CFAP251, LRCC23, and MDH1B by IP-MS and why they identify EFCAB5, AKAP14, and RGS22 by BioID). I believe this reorganization would not remove novelty from the paper but provide robust rationale for the subsequent experiments.

We incorporated the information and atomic model generated by the structural study by Leung et al. (Nature, 2025) in Fig. 1a and 1b, as well as reorganized the whole manuscript. We described this study in the introduction (lines 86 - 92).

2. Even in the absence of describing the recent findings of Leung et al in the introduction, the authors need to describe more accurately what is known about the interactions of CFAP91 from structural information in other species. For example, Walton et al, Nature, 2023 describe the interactions of FAP91 in *C. reinhardtii* flagella and CFAP91 in human respiratory cilia axonemes, including the interaction with CFAP251.

We added a description of CFAP91-CFAP251 interactions found in multiple species in the introduction (lines 80 - 86).

3. The authors write “CFAP91 also showed interactions with RSPH9, which has been discovered to be localized at the RS3 head”. This statement is unfortunately based on inaccurate modeling of low-resolution cryo-ET data of mouse ependymal cilia. Leung and colleagues have demonstrated that the head of RS3 does not contain RSPH9.

We deleted this statement and related results (Fig. 5b) from the manuscript.

4. The work of Leung and colleagues did not report a direct interaction between CFAP91 and LRRC23 in the bovine sperm axoneme (although both exist in the same radial spoke). This discrepancy between results should be discussed.

We discussed it in lines 431 - 433 and deleted related results (Old Fig. 5d).

5. Related to Comments 1-4, the Abstract needs to be rewritten to better represent the current literature.

We rewrote the abstract.

6. The results of the IP experiments are overinterpreted as showing an “interaction” with CFAP91. However, because some of these interactions are likely indirect, a more cautious approach to interpreting these results should be adopted. For example, the authors could replace “interact” with “co-precipitate”.

We agree with the reviewer’s comments. We replaced the words ‘showing interaction’ with ‘found in CFAP91 immunoprecipitates’ throughout the text.

7. P4. L74. The authors state that they have “revealed the function of CFAP91”, which in my opinion does not accurately reflect the conclusions of the work. They have identified potential binding partners of CFAP91 and identified that it is critical for sperm tail assembly, but these points are distinct from identifying the “function” of CFAP91.

We deleted the statement.

8. Videos must be provided showing the movement of spermatozoa from the *Efcab5*^{-/-} and control mice.

We added a video showing the movement of spermatozoa from control and *Efcab5* KO mice, labeled as Supplementary Movie 1.

9. The proteomics data should be deposited in public repositories (e.g. PRIDE).

We deposited our proteomics data to jPOST (JPST003879; PXD065243). We will release it when this manuscript becomes publicly available. In the meantime, our deposition can be viewed using the following link:

<https://repository.jpostdb.org/preview/146768240768550b0ebaca5>; access key: 6347.

Minor comments

1. “Consistently” and “Moreover” are overused in the text, sometimes in consecutive sentences

We reduced the usage of these two words in our manuscript.

2. Do the authors have RT-PCR data for the choroid plexus and oviduct? The Human Protein Atlas indicates that CFAP91 expression is particularly high in these tissues.

We showed the RT-PCR data of *Cfap91* expression in the mouse brain and oviduct in Fig. 1c, which did not show high expression. Additionally, we do not have the RT-PCR data for the mouse choroid plexus.

3. Given the relatively small number of proteins identified by IP-MS, the GO analysis of the CFAP91 interactome (Fig. S5b) could be replaced by manual annotation of the proteins, e.g. IFT-related proteins, radial spoke proteins, other axonemal.

We removed the GO analysis and manually annotated immunoprecipitated proteins (Supplementary Table 2).

4. *C. reinhardtii* is consistently misspelt as *C. reinhartii*

We fixed our spelling.

5. This sentence starting on page 3, line 41 is circular. “indicating that the sperm axoneme is different from the ciliary axoneme” can be removed as the point is already made by the first half of the sentence.

We deleted the redundant part of this sentence.

6. P4. L70. “whether CFAP91 shows the same localization in sperm flagella remained unknown” should be updated in light of recent structural studies.

We incorporated the study by Leung et al. (Nature, 2025) and revised this sentence in lines 86-92.

7. P9. L200. Immunoblot is not verifying the interaction CFAP91 with CFAP251, LRRC23, and IFT140, but simply detecting the presence of these proteins in the immunoprecipitation by a different method

We deleted the words ‘verifying the interaction’ and revised the sentence in lines 245 - 248.

8. P10.L225. hault -> halt

We revised this point in lines 269.

9. P14.L337. The authors state that EFCAB5 was not found in the IP-MS study, but Table S1 suggests otherwise. It was detected with a Log2fold-change of 4.12 and p-value of 0.082.

We intended that EFCAB5 was not significantly enriched in CFAP91 immunoprecipitates. We revised this sentence in lines 469 - 471.

Reviewer #2 (Remarks to the Author):

In this study, Wang et al. investigate the functional role of CFAP91 in spermatogenesis. They demonstrate that CFAP91 interacts with multiple RS3 proteins within the sperm axoneme. Genetic ablation of Cfap91 in mice results in spermatozoa with shortened tails and abnormal head morphology. Through a CFAP91-targeted proximity labeling strategy, the authors identify a sperm-specific RS3 protein, EFCAB5, and show that its depletion impairs sperm motility. While the experimental observations are technically sound, the molecular mechanisms remain inadequately explored. Furthermore, the study does not appear to represent a substantial mechanistic advancement in this field.

Thank you very much for your comments. We performed additional experiments to obtain more molecular mechanisms as written in responses to Major Concerns (1) – (5).

Major Concerns

1) The authors fail to fully elucidate how CFAP91 participates in sperm flagellar biogenesis. Precise spatiotemporal localization of CFAP91 during spermiogenesis may be helpful to distinguish between direct versus indirect roles in axoneme assembly. Although the CFAP91 interactome was characterized in mouse testis, functional validation of its binding partners in flagellar formation remains lacking. Notably, while CFAP91 depletion disrupts CFAP251/LRRC23 localization in flagella, *Lrrc23* knockout mice exhibit normal sperm morphology (PMID: 34585727), suggesting LRRC23 may act downstream of CFAP91. The observed absence of radial spoke proteins in *Cfap91* KO sperm may alternatively result from axonemal structure defects. Thus, critical binding partners directly mediating CFAP91's role in flagellar biogenesis should be experimentally identified.

To carefully examine the localization of CFAP91, we separated testicular germ cells and performed immunocytochemistry. We then found that CFAP91 shows cytoplasmic localization during the round spermatid phase and is transported into the sperm tail during spermiogenesis; we have incorporated this result into Fig. 4e.

Furthermore, in *C. reinhardtii*, a short RS3 is made by the orthologues of CFAP61/91/251 (Walton et al., Nature, 2023. PMID: 37258679). Similarly, in mammals, among the reported mutations of RS3 structural proteins, only CFAP61, CFAP251, and CFAP91 have been shown to impair sperm flagellar biogenesis. Moreover, we found that CFAP91 associates with CFAP251 before the sperm tail starts to form (Fig. 5d), suggesting that CFAP91 and CFAP251 are the units that allow sperm axonemes to elongate. We also demonstrated that CFAP251 could not enter sperm tails without CFAP91 (Fig. 5e). Therefore, our results indicate CFAP251 as the critical binding partner of CFAP91 in sperm flagellar biogenesis. We added these discussions in lines 411 - 423 and 435 - 447.

2) Multiple BBSome components (BBS1, BBS2, BBS4, BBS7) were identified in the CFAP91 interactome. Given established roles of BBSome proteins in flagellar assembly (PMID: 18032602, 15173597, 23572516, 15539463), the authors need to test whether

CFAP91 exerts its effects through BBSome-dependent mechanisms.

As you mentioned, multiple members of the BBS protein family, which forms BBSome, were immunoprecipitated by CFAP91. When we compared the immunoprecipitates of CFAP91 to those of MYCBPAP, an axonemal protein that is essential for sperm tail formation (Wang et al., J. Cell Sci., 2024. PMID: 39092789), BBS proteins were not present in MYCBPAP's immunoprecipitates. Hence, the association with BBSome may be CFAP91-exclusive (Supplementary Fig. 5d). To analyze further the relationship with BBSome, we used an anti-BBS2 antibody, which has been verified to work on immunofluorescence of testicular germ cells (Okitsu et al, Sci. Rep., 2020. PMID: 33144677). Our results indicate that the localization of BBS2 is defective in *Cfap91* KO testes, in which BBS2 forms foci (Supplementary Fig. 5e). This result suggests that CFAP91 may exert its effects through BBSome-dependent mechanisms. We added the results and discussion about BBSome in lines 273 - 284.

3) The sub-Mendelian ratio of *Cfap91*^{-/-} mice in the B6D2 background and potential hydrocephalus phenotypes raise concerns about CFAP91's broader roles in ciliogenesis. The authors should clarify whether CFAP91 depletion affects primary cilia formation in the B6D2 background and determine if its regulatory mechanisms are conserved between ciliogenesis and sperm flagellar biogenesis. Cell-based ciliogenesis models could provide mechanistic insights beyond current *in vivo* approaches.

Hydrocephalus can be caused by abnormal motile cilia or primary cilia. Because CFAP91 is localized in RS3, we hypothesized that CFAP91 deletion may exert its effects by affecting motile cilia. Due to the difficulty in obtaining CFAP91 KO mice in the B6D2 genetic background, we analyzed ependymal and tracheal motile cilia in mice with the B6D2/129 genetic background. Our results showed that motile cilia could elongate without CFAP91 (Supplementary Fig. 3e), in contrast to sperm flagella. We explained the results in lines 161 - 165.

4) While EFCAB5 is identified as a novel RS3 component, and its knockout reduces sperm motility, the molecular mechanisms of EFCAB5 in sperm movement remains unaddressed. The authors propose that "EFCAB5 may regulate sperm-specific motility

patterns by permitting the conformational change of RGS22 which links RS3 to RS2.”
Whether EFCAB5 depletion affects the RGS22 or other RS protein localization?

By performing immunofluorescence using an anti-RGS22 antibody, we found that the localization of RGS22 remained similar in *Efcab5* KO testes (Supplementary Fig. 8a). When we performed immunoblotting for RGS22, CFAP251, and LRRC23 in *Efcab5* KO testes and spermatozoa, these proteins showed similar amounts to their WT counterparts (Fig. 7d). To further investigate the interaction between RGS22 to EFCAB5, we heterologously expressed RGS22 and EFCAB5 in HEK293T cells. Moreover, we analyzed their binding with and without Ca^{2+} , as EFCAB5 possesses a Ca^{2+} -binding domain. Our results showed that EFCAB5 binds to RGS22 independently from Ca^{2+} (Supplementary Fig. 8c). Therefore, the constant connection of RS2 and RS3 through RGS22/EFCAB5 may be important to regulate sperm flagellum motility. We discussed these results in lines 361 - 373.

5) Emerging structural model from bovine sperm flagella (PMID: 39743588) propose domain-specific interactions of CFAP91: its N-terminal region potentially bridges the DMT inner junction, RS2, and N-DRC (implicating roles in axoneme assembly), while the C-terminal domain may recruit motility regulators like EFCAB5. Experimental validation of these predicted interaction domains is essential to unify the study's two major findings.

We expressed the N-terminal or C-terminal half of CFAP91 in HEK293T cells, which were immunoprecipitated with beads. We then mixed beads with testicular lysates to pull down interacting proteins. Our result shows that LRRC23, a motility regulator, immunoprecipitated with the C-terminal half of CFAP91 but not with the N-terminal half. We added the result in lines 423 - 433 and Supplementary Fig. 8d.

Minor Concerns

1) Figures 1a and 6i: Incorporate recent structural models from PMID: 39743588 to enhance schematic accuracy.

We incorporated recent structural models in Fig. 1a and 1b. We also added explanations

of this model in lines 86 - 92.

2) The authors state that “abnormal morphology of Cfap91 KO sperm heads was likely caused by abnormally elongated manchettes.” However, in the discussion part, it is discussed as “this abnormal manchette elongation may be a secondary effect due to impaired flagellum formation”. This contradiction requires resolution.

We intended that CFAP91 is not directly involved in manchette elongation; instead, it could be indirectly involved due to impaired flagellum formation by CFAP91 deletion. We clarified this point in lines 398 - 401.

3) Figure 2a shows apparently normal sperm heads in Cfap9^{-/-} mice, whereas Figure 2c reports near 100% of head abnormalities. Representative images should be re-examined.

We intended to show slight abnormalities on the sperm head hooks; however, it was not clear. We replaced Fig. 2a with clear representative images.

4) Figure 5c, IgG in ^{-/-} TG mice may be a better negative control than ^{+/+} mice.

We performed IP with IgG or anti-LRRC23 antibody and the band of CFAP91-FLAG was only found in the IP product using the LRRC23 antibody. We replaced Fig. 5c and revised sentences in lines 245 - 250.

5) Figure 6b, EFCAB5 signal absence in "Sperm Input" lanes necessitates either longer exposure or alternative blot presentation.

We performed this experiment again and attempted to increase the input concentration; however, EFCAB5 was not found in the input although it was found in the pulldown product. This issue could be caused by low protein abundance and/or low protein solubilization efficiency (0.4% SDS compared to 1% SDS or 6 M urea in other experiments) of EFCAB5 in the input. We explained it in figure legend of Fig. 6b.

Thank you very much for the opportunity to revise our manuscript entitled "Proximity Labeling of Axonemal Protein CFAP91 Identifies EFCAB5 that Regulates Sperm Motility". We sincerely appreciate the reviewers' constructive feedback. We have revised our manuscript in accordance with these comments. We have provided detailed responses (in black) to the reviewers' comments (in blue). The line numbers from the main manuscript without track changes are indicated below.

Reviewer #1 (Remarks to the Author):

In my opinion, the manuscript has undergone considerable improvement since its initial submission. I appreciate the efforts the authors have made to incorporate recent structural data into their analysis of the results. I recommend publication.

We sincerely appreciate your positive feedback.

During reading the revised manuscript, I encountered a few sentences that the authors might want to correct/revise:

L19. It is incorrect to say that radial spokes are "attached to the axoneme" – they are part of it. They are attached to doublet microtubules.

We revised the sentence in lines 23 - 24.

L20. "despite the presence of additional structures on the sperm axoneme" is unnecessary and confusing in the abstract without more information. I would recommend deleting.

We deleted this sentence from the abstract.

L39. It might be more accurate to write "motile cilia" than "the cilia"

We changed 'the cilia' to 'motile cilia' in line 42.

L66. “the underlying mechanism of CFAP91 is not fully understood” – the authors should define what they mean by “mechanism”. Do they mean the mechanism by which CFAP91 regulates ciliary motility? Or the disease mechanism that leads to male infertility?

We explained the ‘mechanism’ that is not fully understood in line 69.

L77. The authors neglect to mention the contribution CFAP91 makes to the N-DRC.

We explained how CFAP91 interacts with N-DRC in lines 80 - 82.

Reviewer #2 (Remarks to the Author):

The authors have addressed my concerns.

We sincerely appreciate your positive feedback.